

# Characterisation of $\delta^{13}CH_4$ source signatures from methane sources in Germany using mobile measurements

Antje Hoheisel[1], Christiane Yeman[1,2], Florian Dinger[1,3], Henrik Eckhardt[1], and Martina Schmidt[1]

[1]Institute of Environmental Physics, Heidelberg University, Heidelberg, Germany
[2]Laboratory of Ion Beam Physics, ETH Zurich, Zurich, Switzerland (now)
[3]Max-Planck Institute for Chemistry, Mainz, Germany

**Correspondence:** Antje Hoheisel (antje.hoheisel@iup.uni-heidelberg.de)

**Abstract.**

The carbon isotopic signature ($\delta^{13}CH_4$) of several methane sources in Germany (around Heidelberg and in North Rhine-Westphalia) were characterised. Therefore, mobile measurements of the plume of $CH_4$ sources are carried out using a cavity ring-down spectrometer (CRDS). To achieve precise results a CRDS analyser, which measures methane ($CH_4$), carbon dioxide

($CO_2$) and their $^{13}C$ to $^{12}C$ ratios, was characterised especially with regard to cross sensitivities of the gas matrix. The two most important gases which affect the measurements are water vapour ($H_2O$) and ethane ($C_2H_6$). To avoid the cross sensitivity with $H_2O$, the air is dried with a nafion dryer during mobile measurements. $C_2H_6$ is abundant in natural gas and thus in methane plumes or samples originating from natural gas. A $C_2H_6$ correction and calibration are essential to obtain accurate $\delta^{13}CH_4$ results, which can deviate up to 3‰ depending on whether an ethane correction is applied.

The isotopic signature is determined with the Miller-Tans approach and the York fitting method. During 21 field campaigns the mean $\delta^{13}CH_4$ values of three dairy farms ($-63.9 \pm 0.9$‰), a biogas plant ($-62.4 \pm 1.2$‰), a landfill ($-58.7 \pm 3.3$‰), a wastewater treatment plant ($-52.5 \pm 1.4$‰), an active deep coal mine ($-56.0 \pm 2.3$‰) and two natural gas storage and gas compressor stations ($-46.1 \pm 0.8$‰) were recorded.

In addition, between December 2016 and June 2018 gas samples from the Heidelberg natural gas distribution network were

measured. Contrary to former measurements between 1991 and 1996 (Levin et al., 1999) no strong seasonal cycle is shown. The mean $\delta^{13}CH_4$ value of this study is $-43.1 \pm 0.8$‰ which is 2.8‰ more depleted than in former years.

## 1 Introduction

Methane ($CH_4$) is the second most important anthropogenic greenhouse gas. The atmospheric growth rate of methane has changed significantly during the last decades, stabilising at zero growth from 1999 to 2006 before beginning to increase again

after 2007 (Kirschke et al., 2013). Several studies have focused on the recent $CH_4$ growth caused by changes in sources and sinks (Rigby et al., 2017; Turner et al., 2017).

Recent studies by Schaefer et al. (2016), Rice et al. (2016) and Nisbet et al. (2016) have shown how the $\delta^{13}CH_4$ measurements can help to understand the changes in global $CH_4$ increase rates, and to assign the related source types. The stable carbon isotope ratio ($^{13}C/^{12}C$) of $CH_4$ sources varies due to the initial source material, and the fractionation during production and



release to the atmosphere. The source categories can be classified as pyrogenic (e.g. biogas burning), biogenic (e.g. wetlands and livestock) or thermogenic (e.g. fossil fuel extraction), which show different but also overlapping isotope ratio ranges. Various studies have shown, that the assignment of isotopic signatures from different $CH_4$ sources remains uncertain due to large temporal variabilities and also regional specificities (e.g. Sherwood et al., 2017). This missing knowledge may result in large

uncertainties when the $CH_4$ budget is determined on global or regional scales using isotope based estimates. In addition to global studies, the use of $\delta^{13}CH_4$ was already successfully applied by Levin et al. (1999) in Heidelberg or Lowry et al. (2001) in London. The study by Levin et al. (1999) showed the $CH_4$ emission reduction in the catchment area of Heidelberg which was accompanied by a significant change in the $\delta^{13}CH_4$ source mixture from $-47.4‰$ in 1992 to $-52.9‰$ in 1995/1996. This was illustrated by decreasing contributions from fossil sources (mainly coal mining).

In order to apply $\delta^{13}CH_4$ in regional models, a better knowledge of the regional source signature of each $CH_4$ source type is needed, taking into account the temporal variations of these sources. For instance, due to the origin the source signature of natural gas in Germany varies between $-55\%$ and $-30\%$ for Russia or North Sea respectively (Levin et al., 1999). In addition to all of these seasonal variations, changes in landfill managements like gas collector systems, and implementation of biogas plants at many farms need to be taken into account for a new study of the global and regional source signature of $CH_4$.

Traditionally, the isotopic ratio of $CH_4$ has been measured with isotope ratio mass spectrometry coupled with GC (Fisher et al., 2006) and this technique is still the most precise, as has been shown by Röckmann et al. (2016) by a comparison of dual isotope mass spectrometry (IRMS), quantum cascade laser absorption spectroscopy (QCLAS), and cavity ring down spectroscopy (CRDS). Instrumental development in measurement technique now allows isotope analysis of $\delta^{13}CH_4$ by CRDS analyser and even its use on a mobile platform (Rella et al., 2015; Lopez et al., 2017). This is a further improvement to the

study of Zazzeri et al. (2015), which involved collecting air samples in bags and analysing them later in the laboratory by IRMS. The studies of Rella et al. (2015) and Assan et al. (2017) demonstrated the importance of a careful determination of cross sensitivities and a good calibration strategy for precise isotope measurements with a CRDS analyser.

In this paper, a strategy to monitor and determine the isotopic carbon source signature of major $CH_4$ sources in Germany using mobile measurements is presented. One major aspect is a careful characterisation of the CRDS analyser to take into

account the cross sensitivity between $\delta^{13}CH_4$ and other components like water vapour and ethane ($C_2H_6$), and to improve the use of a storage tube described by Rella et al. (2015). During 21 mobile measurement campaigns, emission plumes from a biogas plant, three dairy farms, a landfill, a wastewater treatment plant, two natural gas facilities and a bituminous deep coal mine were able to be measured with our setup.

## 2 Methods

### 2.1 Experimental setup

The core component of our experimental setup is the commercially available cavity ring-down spectrometer (CRDS) G2201-i (Picarro, Inc., Santa Clara, CA) which measures routinely the mole fraction of $^{12}CO_2$, $^{13}CO_2$, $^{12}CH_4$, $^{13}CH_4$, and $H_2O$ in an air sample. In addition to the raw spectroscopic measurements, the analyser automatically calculates and outputs the carbon





isotopic signature $\delta^{13}CH_4$ and $\delta^{13}CO_2$. Furthermore, the mole fraction of $C_2H_6$ is measured as an additional feature, which had to be investigated and calibrated for our analyser. A more detailed description of this type of CRDS analyser can be found in Rella et al. (2015). Two different setups are used in this study: a laboratory setup for sample bag analysis and test series and a mobile configuration in a vehicle.

### 2.1.1 Laboratory setup

In the laboratory in Heidelberg the analyser continuously measures ambient air alternating with regular calibration gas and quality control gas injections (Dinger, 2014). In addition, diluted samples from different $CH_4$ sources and gas cylinders can be measured and calibrated. The mobile measurements, using the analyser in a vehicle, are also calibrated using the immediate calibration runs in the laboratory before and after a mobile campaign.

The schematic of the laboratory setup is shown in Figure 1 (a). A 16-port rotary-valve (model: EMT2CSD16UWE, Valco Vici, Switzerland) can be switched automatically by the analyser, to change between different measurements. Ambient air is measured at port 1. Port 3, 7 and 15 are reserved for calibration and quality control measurements. Sample bags are measured on port 11 or 13. The gasflow to the analyser of typically 20 to 80 ml/min is measured by an electronic flow meter (model: 5067-0223, Agilent Technologies, Inc., Santa Clara, CA) before entering the analyser.

Gas samples from different $CH_4$ sources (e.g. natural gas, biogas, landfill gas) need to be diluted, because such samples usually consist of between 50 and 90 % $CH_4$. Therefore, approximately 40 µl of the sample was injected into a three litre bag (Tedlar® with Polypropylen valve with septum, Restek GmbH, Bad Homburg, Germany) filled with synthetic air to dilute the $CH_4$ concentration to approximately 10 ppm in the new sample bag. Due to cross sensitivity with water vapour, these gas samples were dried using a cooling trap below a mole fraction of 0.0015 % water vapour previous to analysis. Each diluted sample was measured for 15 min.

### 2.1.2 Mobile measurement setup

For mobile measurements the CRDS analyser is installed inside a vehicle and measures air while driving. The system consists of the CRDS analyser, a nafion dryer and a storage tube, the so called AirCore (Karion et al., 2010), which enable us to remeasure the stored air from the last 2 minutes of continuous measurement (Fig.1 (b)). The AirCore was built after Rella et al. (2015) using a 25 m Decarbon tubing with an inner diameter of 9.5 mm and a volume of 1.77 l (Yeman, 2015). The setup of the nafion dryer is similar to the one built by Welp et al. (2013) using a Perma Pure MD-070-96-S nafion dryer and a vacuum pump. The CRDS device and the vacuum pumps are powered by a portable power source (260 h deep cycle battery (Winnerbatterien Germany) and a 1000 W inverter which offers 230 V output) which allows for over 12 hours of measurement time.

The ambient air enters the air intake line 20 cm above the vehicle roof. It can follow two different paths to the analyser depending on the valve positions. In the 'monitoring mode', indicated by blue arrows in Fig.1 (b), the ambient air enters the CRDS analyser after the air is dried with the nafion dryer to a mole fraction of less than 0.1 % water vapour. Simultaneously, a second split-off flow leads the ambient air through the AirCore. Due to the length of the intake line, the volume of the cavity, and a flow rate of 0.16 l/min the air needs approximately 20 to 25 sec to be measured in the CRDS analyser.



The vehicle usually passes an emission plume of a $CH_4$ source within 40 sec and the analyser records approximately 10 data points per $CH_4$ peak. To achieve higher time resolution and accuracy for $\delta^{13}CH_4$ analysis, it is possible to remeasure $CH_4$ peaks by analysing the air stored in the AirCore with the 'replay mode'. This enables us to remeasure the stored air that contains the sampled $CH_4$ peak. The average analysis time is then 4.5 min corresponding to approximately 70 data points and

thus the measurement in 'replay mode' has a higher time resolution and a better precision than the one in 'monitoring mode'.

In Figure 2 (left) a typical mobile measurement of a plume from a biogas plant close to Heidelberg (Germany) is shown for $CH_4$ concentrations and $\delta^{13}CH_4$ values. The vertical black line indicates the switching from 'monitoring mode' to 'replay mode'. The small dots represent the reported data in monitoring (blue) and replay (black) mode, logged approximately every 3.7 sec, while the red lines show the 15 sec averages in 'replay mode'. For comparison the peak measured in 'monitoring

mode' (blue dots/line on the left side) is stretched by a factor of 12.5 (blue line on the right side) so that the representations of both peaks have the same width. The peak measured in 'replay mode' precisely corresponds to the stretched one measured in 'monitoring mode', because both peaks reproduce the same emission plume. This differs from the AirCore measurements performed by Lopez et al. (2017) which show higher $CH_4$ values in replay than in 'monitoring mode'.

During the mobile measurements the vehicle position was recorded by a GPS mouse (Navilock 602u) with an accuracy of

2 m CEP (circular error probable). A weather station (Vantage Pro2 ™, Davis Instruments) was set up near the measurement site to record the wind speed and direction, the temperature and the incident solar radiation.

## 2.2 Characterisation of the CRDS analyser G2201-i

### 2.2.1 Correcting the measured $\delta^{13}CH_4$ values

With regard to the publications of Rella et al. (2015) and Assan et al. (2017) our main focus during the instrumental charac-

terisation was on $\delta^{13}CH_4$. The cross sensitivities of $H_2O$, $CH_4$, $CO_2$ and $C_2H_6$ concentrations on $\delta^{13}CH_4$ were investigated to determine correction factors. The correction factors subsequently applied in this study are summarised in Table 1. The correction and calibration scheme is sketched in Fig. 3 and described in more detail in Hoheisel (2017). The $H_2O$ interference on $\delta^{13}CH_4$ was tested by carrying out several humidity tests (Fig. S1). For this purpose, two dry compressed air gases with gas mixtures of 2.3 ppm and 10.1 ppm $CH_4$ were humidified by flushing them through a reversed glass condensation trap kept at

room temperature and filled with one droplet of deionised water. Due to evaporation of the water droplet the humidity of the gas passing the condensation trap changed with time between 1.5 to 0 % water vapour. Rella et al. (2015) recommended a reduction of the humidity below a mole fraction of 0.1 % water vapour for accurate $\delta^{13}CH_4$ results. Our tests confirm this recommendation for humidity levels below 0.15 % but observed a significant cross sensitivity of $0.54 \pm 0.29 \, (\permil \, ^{13}\delta CH_4)(\% \, H_2O)^{-1}$ for humidity levels above 0.15 %. To reduce possible uncertainties due to humidity correction, the air was dried with a nafion dryer

below a mole fraction of 0.1 % water vapour during mobile measurement. However, the nafion drying unit was not installed until September 2016, so the measurements before this date were corrected.





Additionally, the cross sensitivities of $CH_4$ and $CO_2$ on $\delta^{13}CH_4$ were tested (Fig. S2 and S3). Two dilution tests were carried out, generating different gas mixtures. No significant cross sensitivities of $CH_4$ and $CO_2$ on $\delta^{13}CH_4$ were detected up to concentrations of 10 ppm $CH_4$ or rather 450 ppm $CO_2$.

Previous studies from Rella et al. (2015) and Assan et al. (2017) have reported higher $\delta^{13}CH_4$ results when the gas sample
contains $C_2H_6$. As natural gas contains between 1.4 to 7 Mol% of $C_2H_6$ (Nitschke-Kowsky et al., 2012), the $C_2H_6$ interference is especially important when analysing $CH_4$ emissions from natural gas facilities or the isotopic composition of natural gas. The $C_2H_6$ interference on $\delta^{13}CH_4$ measurements was carefully tested with our analyser by carrying out three dilution tests, to determine a correction (Fig. S4). $\delta^{13}CH_4$ increases linearly with increasing $C_2H_6$ to $CH_4$ ratio. The slope of the regression line and thus the correction factor was found to be $40.87\pm0.49‰\,(ppm\,CH_4)/(ppm\,C_2H_6)$. The correction is necessary due
to $\delta^{13}CH_4$ showing a bias of up to 3‰ in our study depending on the $CH_4$ to $C_2H_6$ ratio of the sample and the calibration cylinder.

### 2.2.2   Correcting the measured $C_2H_6$ concentration

To correct for the strong cross sensitivity between $C_2H_6$ and $\delta^{13}CH_4$ measurements, an accurate determination of the $C_2H_6$ concentration is required. Because the measurement of $C_2H_6$ is an additional feature of the instrument a correction and cali-
bration of the $C_2H_6$ concentration were performed.

The $C_2H_6$ concentration decreases strongly with increasing humidity, even for $H_2O$ concentrations below 0.15 % (Fig. S1). For humidity below 0.15 % a correction factor of $0.43\pm0.51\,(ppm\,C_2H_6)/(\%\,H_2O)$ was determined and for humidity higher than 0.16 % the correction factor is $0.70\pm0.10\,(ppm\,C_2H_6)/(\%\,H_2O)$. There is no correction for $H_2O$ mole fractions between 0.15 and 0.16 %, because in this range the behaviour of $C_2H_6$ in the presence of $H_2O$ changes. However, no discontinuity, such
that observed by Assan et al. (2017), was seen.

Besides $H_2O$ also the concentrations of $CH_4$ and $CO_2$ interfere with the measured $C_2H_6$. To determine the cross sensitivities of $CH_4$ and $CO_2$ on $C_2H_6$ two dilution series and three injection tests were performed and produced gas mixtures with concentration ranges of 1.8 to 10 ppm $CH_4$ or 2 to 600 ppm $CO_2$. All dilution and injection tests with $C_2H_6$ concentrations between 0 to 1.3 ppm show similar results with an average of $0.0077\pm0.0007\,(ppm\,C_2H_6)/(ppm\,CH_4)$ and $(1.25\pm0.94)\,10^{-4}\,(ppm$
$C_2H_6)/(ppm\,CO_2)$ (Fig. S5).

To calibrate the $C_2H_6$ measurement two dilution tests with $C_2H_6$ concentrations ranging from 0 to 3 ppm were performed (Fig. S6). The measured $C_2H_6$ concentrations were nearly twice as large as expected. After correcting the measured $C_2H_6$ concentrations due to $H_2O$, $CH_4$ and $CO_2$ a calibration factor (slope of the regression line) of $0.538\pm0.002$ ppm/ppm and a calibration intercept of $0.070\pm0.005$ ppm was determined.

### 2.2.3   Calibration to international scales

All calibration gases used in this study are compressed air filled in aluminium cylinders. The $CH_4$ and $CO_2$ concentrations were calibrated against the WMO scale (Dlugokencky et al., 2005) using a GC system (Levin et al., 1999). To determine the $\delta^{13}CH_4$ values, flasks filled from our calibration gases were sent to MPI Jena ($\delta^{13}CH_4$: $\pm0.05$ ppm). These analyses connect





our Heidelberg measurements to the VPDB (Vienna Pee Dee Belemnite) isotope scale (Sperlich et al., 2016). $C_2H_6$ is not fully calibrated to an international scale. One calibration cylinder filled by Deuste-Steininger (Mühlhausen, Germany) with 4.98 ppm $C_2H_6$ is certified by this company with an uncertainty of $\pm 2\%$.

All data measured with the CRDS analyser in the laboratory or during mobile campaigns was corrected prior to the one point
calibration calculation using the factors from Table 1 and following Fig. 3. The gas cylinder used for calibration was chosen according to the experiment to ensure a similar composition and concentration range for sample and standard. For ambient air measurements in the laboratory and for mobile measurements a gas cylinder filled with compressed air is used to calibrate the data. For diluted gas samples from $CH_4$ sources a gas cylinder with atmospheric concentrations spiked with natural gas to 10 ppm $CH_4$ is used. The calibration gas is measured before and after every experiment/field campaign in the laboratory or in the
vehicle. Tests at the beginning of this study showed that measurements of the calibration gas inside the vehicle do not increase the precision and are therefore not necessary for mobile measurements of less than 10 hours.

### 2.2.4   Instrument performance and uncertainties

The repeatability of the analyser as a function of the $CH_4$ concentration was determined by the measurement of three different gas cylinders for 120 min each. The Allan variance (Werle et al., 1993) was calculated with the raw data for averaging times
of up to 11 min (Fig. 4). The Allan standard deviation $\sigma$ (the square root of the Allan variance $\sigma^2$) for the raw (3.7 sec) $CH_4$ data is between 0.34 to 2.69 ppb for gases with a $CH_4$ concentration of 1900 to 10000 ppb. For the corresponding $\delta^{13}CH_4$ data, an improvement of the Allan standard deviation with higher $CH_4$ concentration from 3.76 to 0.77‰ can be seen. The Allan standard deviation of $C_2H_6$ is approximately 0.09 ppm for gases with $C_2H_6$ concentrations up to 5 ppm .

During mobile measurements especially $CH_4$ and $\delta^{13}CH_4$ show rapid changes when driving through the emission plume
of a $CH_4$ source and thus do not allow us to average the data over long time periods. However, for sample measurements in the laboratory (e.g. natural gas samples) longer averaging times of up to 10 or 15 min significantly decrease the Allan standard deviation (see Fig. 4). For a 10 min averaging period the Allan standard deviation of 1900 ppb or 10000 ppb $CH_4$ decreases to values of 0.09 ppb and 0.47 ppb, and for $\delta^{13}CH_4$ to values of 0.40‰ and 0.06‰. The Allan standard deviation of $C_2H_6$ decreases to 0.006 ppm. Due to the correction and calibration of $\delta^{13}CH_4$ there is a relative increase in the uncertainty of
approximately 5 to 12 %.

## 2.3   Analysis of $\delta^{13}CH_4$

### 2.3.1   Gas samples from natural gas distribution network

Between December 2016 and June 2018, gas samples from the Heidelberg natural gas distribution network were collected two to three times a month from the gas blowing workshop at the university campus in one litre sample bags (Tedlar® with
Polypropylen valve with septum, Restek GmbH, Bad Homburg, Germany).

The gas samples were measured as described in Sect. 2.1.1, corrected by the factors given in Table 1 and calibrated as described above. For each gas sample the average and standard deviation of the 10 min measurement were calculated.





To determine the repeatability of a measurement as well as the storage effect, pair samples were taken and storage tests carried out, with storage times of the bags up to 226 days and two to five measurements taken from each sample bag. Duplicate samples taken on the same day and measured one after another show a mean difference in $\delta^{13}CH_4$ of $0.12 \pm 0.08‰$ with a maximal difference of $0.30‰$. Storage tests of 12 natural gas samples stored on average for 104 days (41 to 226 days) in

Tedlar® bags show an average drift of $0.0023 \pm 0.0028‰/day$. Since the samples are measured for the first time on average 26 days (0 to 88 days) after the sample day, the $\delta^{13}CH_4$ signature of the samples will change by approximately $0.06‰$ due to this storage in Tedlar® bags. Even after 100 days the average drift is only $0.23‰$ and therefore for each sample the $\delta^{13}CH_4$ values measured within 100 days after sampling were averaged. To quantify the variations of $\delta^{13}CH_4$ from the local gas supply network within one week, two samples per day were taken over 5 days at the end of November 2017 and averaged the $\delta^{13}CH_4$

values for the duplicate samples. The maximal difference between the five averaged values was $0.7 \pm 0.2‰$.

### 2.3.2  Determination of $\delta^{13}CH_4$ source signatures from mobile plume measurements

For mobile measurements the CRDS analyser is installed inside a vehicle and measurements are carried out as described in Sect. 2.1.2. The $\delta^{13}CH_4$ signature of the $CH_4$ sources were determined by the Miller-Tans approach (Miller and Tans, 2003) using the unaveraged data measured in 'replay mode' with the AirCore. To fit a linear regression line to the data the York fit

(York et al., 2004) was used as recommended also by Wehr and Saleska (2017). Because the York fit allows errors in x and y, it also account for the finding that the analyser can measure $\delta^{13}CH_4$ more accurately at higher $CH_4$ concentrations. The errors for $CH_4$ and $\delta^{13}CH_4$ for different concentrations were determined with the Allan standard deviation.

For accurate results the following criteria are used to select 79 AirCore measurements out of 135. Only $\delta^{13}CH_4$ signatures with uncertainties lower than $5‰$ are used. The number of data points and especially the peak height above background

concentration control the precision of the determined isotopic signature when applying a Miller-Tans Plot, therefore only plume measurements with peak heights above background concentration higher than 0.45 ppm and more than 25 data points fulfil this criterion. Furthermore, in some cases the reported $C_2H_6$ concentration jumps while driving although there cannot be a change in the $C_2H_6$ concentration of the ambient air. These jumps in $C_2H_6$ also results in $\delta^{13}CH_4$ jumps. Therefore, all AirCore measurements with a sudden change in $C_2H_6$ larger than 1 ppm were neglected. With these criteria the isotopic

signature of a $CH_4$ source determined from one AirCore plume measurement has an average precision of $1.8 \pm 1.3‰$.

### 2.3.3  Comparison of different methods to determine $\delta^{13}CH_4$ source signatures

In order to define the optimal method for the determination of the source signature the 135 AirCore measurements as well as simulated data were used. In the following the differences in the $\delta^{13}CH_4$ source signature when using the Keeling method or the Miller-Tans approach (Keeling, 1958; Miller and Tans, 2003) will be discussed and the York fit will be compared to the

ordinary least squares (OLS) fit (here the lm() fit function from GNU R is used).

Similar to the method described by Wehr and Saleska (2017) for $CO_2$ and $\delta^{13}CO_2$, we simulated several typical emission plume crossings with $CH_4$ source signatures of $-35‰$ to $-65‰$ and a background of $-48‰$. In addition, the $CH_4$ concentration enhancements in the plume $\Delta c_{source}$ (100 − 10000 ppb), the number of measured data points during plume crossing n





$(10 - 280)$ and the averaging times (up to 1 min) were varied. For each set of conditions ($\delta^{13}CH_{4source}$, $\Delta c_{source}$, n), we generated synthetic $CH_4$ concentrations and calculated the corresponding $\delta^{13}CH_4$ values using a background concentration of 1.95 ppm $CH_4$ and a Gaussian curve with n equidistant data points every 3.7 s and a peak height of $\Delta c_{source}$. To reproduce the statistical uncertainties of a real measurement, we add a normally distributed scattering around zero to the synthetic $CH_4$

concentrations and the corresponding isotope ratios. The standard deviation of the normal distributed scattering depends on the $CH_4$ concentrations and was chosen as the Allan standard deviation measured for raw data of the analyser. However, when simulating possible improved analysers we reduced the scattering by a factor 2 to 10. Such sets of data were generated 5000 times for each condition. For each dataset the $\delta^{13}CH_4$ source signature was calculated with the Miller-Tans and the Keeling method using the York or the OLS fit.

For the York fit the $\delta^{13}CH_4$ source signature determined using the Miller-Tans approach is identical within the relevant order of magnitude to the one calculated using the Keeling method. This can be shown for the AirCore measurements and is confirmed by our simulations. Figure 2 (right) shows an example of the Keeling Plot (upper panel) and the Miller-Tans Plot (lower panel) used to calculate the isotopic signature of the corresponding $CH_4$ source.

   The $\delta^{13}CH_4$ signature calculated with the simple OLS fit out of the AirCore measurements differ between $-2$ and $2 \permil$

depending on the method which is used (Miller-Tans or Keeling method). This finding is in agreement with the simulated results.

   Comparing the measured isotopic signatures of $CH_4$ resulting from York and OLS fit in approximately 90 % of the measurements the result of the York fit lies in between the results from the OLS with the Miller-Tans and the Keeling method. This agrees well with our simulated results, where the value of the $\delta^{13}CH_4$ signature determined with the York fit for peak

enhancements between 0.1 and 3 ppm lies between the values calculated for the OLS fit with Miller-Tans and Keeling method in more than 98.5 % of the results.

   The average values for the 5000 determined isotopic signatures for the York and the OLS (Keeling and Miller-Tans) fit in this study are nearly the same ($< 0.05 \permil$ for $CH_4$ ranges higher than 0.2 ppm) and have in all three cases significant larger differences to the true value ($< 0.2 \permil$ for $CH_4$ ranges higher than 0.2 ppm and $< 0.1 \permil$ for $CH_4$ ranges higher than 0.6 ppm)

than between each other. However, the 5000 individual simulated values for the $\delta^{13}CH_4$ signatures for one condition vary widely around the average and the true value.

   Due to the above described comparisons, the York fit and the Miller-Tans approach were chosen to determine the $\delta^{13}CH_4$ source signature in our study. A further characterisation of this method showed that the uncertainty of a single source signature determination depends mainly on three criteria: the $CH_4$ range, the number of data points used for the fit and the precision of

the analyser.

   The first large limitation for a precise determination of the isotopic source signature is the $CH_4$ concentration of the plume above background. The higher the $CH_4$ peak the more accurately the $\delta^{13}CH_4$ signature can be determined. Especially for small $CH_4$ sources, it is important to drive as close as possible to the source to increase the peak height. In Fig. 5 the uncertainty of the isotopic signature of every AirCore measurement (black dots) is given as a function of $CH_4$ peak height above background.

For $CH_4$ enhancements lower than 1 ppm the uncertainty increases strongly to values higher than $20 \permil$. The coloured lines



show the standard deviation of the 5000 synthetic data with different numbers of data points used for the Miller-Tans approach. The synthetic data agrees well with the measured values which were calculated out of 25 to 280 data points.

The second parameter which influences the accuracy of the determined $\delta^{13}CH_4$ signature is the amount of data points. During measurements significantly different isotopic signatures were measured in 'monitoring' (approximately 10 data points)

and in 'replay mode' with the AirCore (on average approximately 70 data points) (see Fig. 2). The synthetic data confirms that with increasing amount of data points the uncertainty of the $\delta^{13}CH_4$ signature improves (Fig. 5). The precision can be more than doubled by increasing the number of points from 10 to 70 and more than quadrupled by an increase from 10 to 280. In 'monitoring mode' the amount of data points per peak is constrained by the small width of the plume and the driving speed. Therefore, it is important to remeasure the plume using the AirCore to increase the number of data points and thus the

precision.

The third limitation of the accuracy of the determined source signature is the measurement precision of the instrument for raw (3.7 sec) data, especially for $\delta^{13}CH_4$. The measuring intervals of the plume are short and thus the $CH_4$ concentration and isotopic signature change rapidly, making it impossible to increase the precision through averaging over time periods longer than one minute. The value as well as the uncertainty of the isotopic signatures determined from the original and the

15 sec averaged data from AirCore measurements do not show significant differences using the Keeling or the Miller-Tans approach for the real measurements (see Fig. 2). Moreover, additional tests with synthetic data show that averaging over 7 to 60 sec improves the precision of the measurement, but not the source signature determination due to a smaller amount of data points. Therefore, the raw unaveraged data from the analyser measured in 'replay mode' was used instead of the averaged one. The Allan standard variance without averaging for $\delta^{13}CH_4$ is up to 3.76‰ (1.9 ppm $CH_4$). An increase of the precision to a

standard deviation of 1‰ would lead to a nearly four times better precision of the determined isotopic source signature. For future measurements more precise instruments are important. Finally, simulated results for different isotopic source signatures were compared and no dependence on the determined methane source signature was noticed.

# 3   Results

## 3.1   $\delta^{13}CH_4$ from Heidelberg gas distribution network

Between 1991 and 1996 measurements of the natural gas distribution network in Heidelberg were carried out by Glatzel-Mattheier (1997). The measured $\delta^{13}CH_4$ signatures underlied a strong seasonal variation with $-30$‰ in winter and up to $-50$‰ in summer. The annual average was $-40.3 \pm 3.0$‰ (Glatzel-Mattheier, 1997; Levin et al., 1999). The seasonal cycle in the isotopic signature in the 1990s was explained by seasonal changes in gas imports with a larger contribution from Russian gas in summer months and mainly from northern Germany and Scandinavia during winter, because the isotopic signature

of natural gas differs depending on its formation process and therefore its origin. Natural gas from Siberia has an isotopic signature between $-48$ to $-54$‰ (Cramer et al., 1998) and is thus less enriched than North Sea gas with $\delta^{13}CH_4$ values of approximately $-34 \pm 3$‰ (Lowry et al., 2001). In the late 1990s the percentage of natural gas from import and domestic





production in Germany (BAFA, 2017) varies with the seasons. While in summer 1998 and 1999 approximately 44 % of the natural gas imports in Germany originate from Russia, in winter it was only 25 to 30 %.

Between December 2016 and June 2018 the measured $\delta^{13}CH_4$ signatures vary between $-44.7$‰ and $-41.4$‰ with an average value of $-43.1 \pm 0.8$‰ (Fig.6). No significant seasonal cycle has been observed during these 19 months. The mea-
surements in our recent study show that natural gas in Heidelberg is nowadays on average approximately 2.8‰ more depleted than in the 1990s. The percentage of natural gas from import and domestic production in Germany (BAFA, 2017) affirm our findings of no significant seasonal cycle, with reporting a mixture of natural gas which is nearly the same over the year. It should be noted that the statistics are for Germany as a whole, while no information for the Heidelberg region is available from the local gas network.

A closer look at the measured isotopic signature of natural gas of the last year (2017) (Fig.6) shows that the isotopic signature of natural gas in Heidelberg is more depleted in winter than in summer, which is opposite to the results found in the 1990s. Our results can indicate that the percentage of Russian gas is higher in winter than in summer. The $C_2H_6$ to $CH_4$ ratio seems to support this trend with lower values in winter, 0.04, and higher ones in summer, 0.06. Nitschke-Kowsky et al. (2012) reported the $C_2H_6$ to $CH_4$ ratio for Russian natural gas to be 0.014 while for North Sea gas it is 0.078.

## 15   3.2   $\delta^{13}CH_4$ source signatures from mobile measurements

The $\delta^{13}CH_4$ signature for different methane sources (see Fig.7) are determined out of 135 plumes measured over 21 days while using the AirCore. For the evaluation only 79 AirCore measurements with peak heights of more than 0.45 ppm above background and more than 25 data points were selected (see Sect. 2.3.2). During each measurement day one to five AirCore measurements were carried out at selected $CH_4$ sources and the determined isotopic signatures of each $CH_4$ source were
averaged to a daily mean (see Fig. 8, Table 2, and Supplement Table S1).

In the following the determined isotopic signatures of $CH_4$ sources will be discussed for every measuring site and be compared with values from other studies and $\delta^{13}CH_4$ signatures measured from gas samples taken at selected measuring sites.

### 3.2.1   Biogas plant

In biogas plants, microbial organisms produce $CH_4$ under anaerobic conditions. The isotopic signature of $CH_4$ in biogas can vary widely due to the substrate, the microbial producers of $CH_4$ and kinetic values like temperature and frequency of feeding (Polag et al., 2015; personal communication with D. Polag, 2017).

The biogas plant Pfistererhof in Heidelberg has two fermenter tanks. One is fed with a substrate mainly consisting of maize silage and the other predominantly of food waste. Gas samples from both fermenter tanks were taken and measured. The
$\delta^{13}CH_4$ signature of the produced biogas was $-61.5 \pm 0.1$‰ for the maize-silage tank and $-64.1 \pm 0.3$‰ for the food-waste tank. Therefore, the isotopic source signature determined out of the measurement of the $CH_4$ plume is expected to lie between the above mentioned values, because $CH_4$ from both fermenter tanks is mixed downwind of the biogas plant.





Over 10 days, mobile measurements were carried out downwind of the biogas plant between August and December 2016 and in February and March 2017. The maximum $CH_4$ peak height of the measured plumes varied between 2.5 and 17 ppm. Often multiple peaks were measured while driving through the plume, caused by several sources on the biogas plant. The isotopic signatures of $CH_4$ emitted by the biogas plant were determined out of 17 measured plumes. The values varied between $-59.0$ and $-64.2‰$ with one exception of $-67.4‰$ and the overall average of $\delta^{13}CH_4$ was $-62.4 \pm 1.2‰$. The overall average and also the daily averages agree well with the isotopic signatures of the direct samples.

### 3.2.2 Dairy farms

The $\delta^{13}CH_4$ source signature emitted at three dairy farms (in Ladenburg, Weinheim and Kleve) were characterised. The dairy farm in Weinheim holds 320 to 340 dairy cows and the one in Ladenburg holds 80 dairy cows. Haus Riswick in Kleve is an education and research centre of the Agricultural Chamber of North-Rhine Westphalia with 230 dairy cows in conventional livestock farming, 45 dairy cows in organic livestock farming and more than 200 sheep and calves each. In the largest dairy cowshed in Kleve (conventional dairy cowshed) feeding experiments and emission measurements have been carried out (Schiefler, 2013; Schmithausen et al., 2016).

All three dairy farms have an associated biogas plant. This is not representative for Germany because most dairy farms do not have such a facility. In 2013 there were 285 000 agricultural holdings in Germany, 45.8 % of them were cattle farms including dairy cow farms. But only 2.2 % (6 300) of all agricultural holdings had a biogas plant and thus much less than 5 % of all cattle farms (including dairy cow farms) could have a biogas plant (Agrarstrukturerhebung, 2013).

Levin et al. (1993) showed that the isotopic signature of $CH_4$ produced by cows strongly depends on the diet. Cows with a 100 % C3 ($-65.1 \pm 1.7‰$) diet emit less enriched $CH_4$ than cows with a 60 to 80 % C4 diet ($-55.6 \pm 1.4‰$). In addition, $CH_4$ emitted by liquid manure has a more depleted isotope ratio of $-73.9 \pm 0.7‰$.

The dairy cows in Weinheim are full-time in the cowsheds and were fed nearly identically throughout the year with 36 % C4 plants (maize) and 64 % C3 plants. Therefore, no strong variations in the determined $\delta^{13}CH_4$ signature of $CH_4$ would be expected. However, the values vary between $-40$ to $-66‰$. A more detailed inspection of the origin of the peaks showed a possible influence of the biogas plants placed on the farms. In Ladenburg and Weinheim most wind conditions made it impossible to separate between $CH_4$ produced from the cows and from the biogas plant. To determine the $CH_4$ emissions from the dairy cows and the cowshed only AirCore measurements with distinct wind directions were used. These measurements were carried out directly next to the cowshed on the farm, where an influence of the biogas plant could be excluded.

In Weinheim only 3 out of 15 plume measurements were used (Sep 16 to Feb 17), because during all other samplings an influence of the biogas plant cannot be excluded. These AirCore measurements were taken when driving directly over the farm. Therefore, the peak heights were relatively high with 8.3 and 8.9 ppm. The $\delta^{13}CH_4$ values varied between $-62.6$ and $-66.0‰$ with an average of $-64.9 \pm 1.6‰$. For the ten other AirCore measurements $CH_4$ emitted from cowshed and biogas plant cannot be separated. The resulting mean isotopic signature is $-54.0 \pm 8.0‰$ spanning a range between $-43.1$ and $-62.6‰$. The plumes measured downwind of the dairy farm had peak heights between 2.6 and 9 ppm with an average of 4.3 ppm.





Next to the dairy farm in Ladenburg the plumes measured over 6 days between October 2016 and February 2017 had most of the time very small peak heights of 2.1 to 2.8 ppm (on average 2.4 ppm). As expected the plumes were smaller than the ones measured near the dairy farm in Weinheim although the measurements were carried out closer to the source. Only on one day in November 2016 a $CH_4$ concentration of up to 8 ppm was measured in the plume. The $\delta^{13}CH_4$ signatures determined

out of three AirCore measurements taken when driving on the road next to the farm have values around $-44.4 \pm 0.8$‰. For these measurements it was not possible to separate between $CH_4$ emitted by the cows and by the biogas plant. To determine the isotopic signature of $CH_4$ from the dairy cows and the cowshed alone, three AirCore measurements of the plume directly on the farm next to the cowshed were taken in October 2016, which had concentrations up to 4.1 to 7.3 ppm. The determined $\delta^{13}CH_4$ values varied between $-61.6$ and $-64.0$‰ with an average of $-63.2 \pm 1.4$‰.

In Weinheim as well as in Ladenburg the $\delta^{13}CH_4$ signature of the whole farm (cowshed and biogas plant) is less depleted than the isotopic signature of the cowshed alone. Further experiments are needed to determine the seasonal isotopic signature of the biogas plants on dairy farms and the influence on the plume of the farm in total.

On 24 March 2017 five AirCore plume measurements were taken on the dairy farm in Kleve with maximal $CH_4$ concentrations between 4.7 and 13.6 ppm. The determined $\delta^{13}CH_4$ signatures vary between $-61.7$ and $-65.1$‰ and the average

is $-63.5 \pm 1.6$‰. The weather conditions made it possible to exclude an influence from the biogas plant. Two measurements were taken directly next to both the large cowshed with dairy cows of conventional farming and next to the cowsheds of organic keeping. The average isotopic signatures of $CH_4$ emitted by the cowsheds of conventional and organic livestock farming do not differ significantly. For conventional livestock the determined $\delta^{13}CH_4$ signature is $-64.3 \pm 1.5$‰ and for organic livestock $-64.4 \pm 0.9$‰. The fifth AirCore measurement was done on the downwind side of the farm ($-61.7 \pm 1.7$‰).

The average $\delta^{13}CH_4$ signature of all three dairy farms match each other and the isotopic signature expected from the results from Levin et al. (1993). It is important to note that the measured $CH_4$ from the plume of cowsheds is a mixture of $CH_4$ emitted by cows and manure.

### 3.2.3   Landfill

Bergamaschi et al. (1998) determined $\delta^{13}CH_4$ signatures of different sample types from four German and Dutch landfills.

For direct gas samples from the gas collecting system they measured an isotopic signature of $-59.0 \pm 2.2$‰. Emission samples taken with static chambers at covered areas of the landfill showed, however, more enriched isotopic signatures of $-45.9 \pm 8.0$‰. Due to the presence of oxygen in the upper soil layers, aerobic bacteria oxidate parts of $CH_4$ which diffuses through the soil cover and shift the isotopic signature to higher values. Upwind-downwind measurements of $CH_4$ around the landfill lead to an isotopic signature of $-55.4 \pm 1.4$‰.

In this study, the isotopic signature of $CH_4$ emitted from a landfill with a disposal area of approximately $1.45 \, km^2$ which is located near Sinsheim, south-east of Heidelberg, was characterised. From 1978 to 1998 biodegradable domestic waste was deposited there. A degassing system collects the produced biogas which is used to generate electricity (AVR, 2016). The landfill is covered in large parts by a final surface sealing and during the measuring period construction works were done to cover further parts.




Over 10 days from July to November 2016 and in March and July 2017, 26 plume measurements were performed. During this period the $CH_4$ plume was measured twice on the landfill and the other times while driving on a public road next to it. The measured $CH_4$ concentrations of the plumes downwind of the landfill were relatively small with 2.1 to 2.7 ppm. Therefore, the $\delta^{13}CH_4$ signature cannot be determined to a high accuracy. From 18 measured plumes only four can be used to determine

the isotopic source signature precisely. The resulting values vary between $-54.2$ and $-62.2$‰. No seasonal variations has been observed. The average daily mean is $-58.7 \pm 3.3$‰. This result is comparable to the upwind-downwind measurements of $CH_4$ by Bergamaschi et al. (1998) and to the study of Zazzeri et al. (2015) in the UK with values between $-55.2 \pm 0.6$‰ and $-60.2 \pm 1.4$‰, with an average of $-58.0 \pm 3.0 (2SD)$‰.

In July 2016 the $CH_4$ concentration was measured directly on the landfill. The maximum measured concentration was, with

values up to 6 ppm, higher than the ones measured downwind of the landfill. The average $\delta^{13}CH_4$ signature is $-66.5 \pm 2.5$‰ ($-64.0$ to $-69.3$‰). Nearly one year later measurements were carried out on the landfill again. The average $\delta^{13}CH_4$ signature out of two AirCore measurements is, with $-59.5 \pm 0.5$‰ ($-59.9$ and $-59.1$‰), much more enriched and in good agreement with the measurements next to the landfill. Again the $CH_4$ peaks were, with values between 2.6 to 7.2 ppm, higher than the measurements downwind of the landfill.

Direct gas samples from the gas collecting system taken on the same day in July 2017 have an average isotopic signature of $CH_4$ of $-59.5 \pm 0.1$‰. This value matches the isotope ratio of $-59.0 \pm 2.2$‰ reported by Bergamaschi et al. (1998) for direct samples from the gas collecting system. Like Bergamaschi et al. (1998) the isotopic signature of $CH_4$ in the gas collecting system is less enriched than the isotope ratio measured in the plume next to the landfill. The isotopic signature of $CH_4$ determined out of the plume on the landfill in July 2017 is the same as for the direct gas sample. The large $CH_4$ peaks measured on the

landfill seem to originate from the gas collecting system.

As previously mentioned, less enriched $\delta^{13}CH_4$ values of $-66$‰ were determined out of measurements carried out on the landfill in July 2016. Bergamaschi et al. (1998) measured such depleted $\delta^{13}CH_4$ signatures of approximately $-69$‰, too, once for a gas sample from the gas collecting system and in one depth profile measurement. Our measurement may have been influenced by constructions work which were done on the landfill during the whole measurement period.

### 3.2.4 Wastewater treatment plant

Every year approximately 23 million $m^3$ of wastewater is cleaned in the wastewater treatment plant (WWTP) in Heidelberg. During our field campaigns mobile measurements were carried out next to the southern part. There, the sludge treatment inside the digestion towers takes place in three septic tanks with a volume of 2500 $m^3$ each. The produced sewage gas consists predominantly of $CH_4$ and is collected to be utilised in a block heating station (Abwasserzweckverband Heidelberg, 2017).

In February 2017 two gas samples of the collected gas were taken from the WWTP and were analysed in the laboratory. The average $\delta^{13}CH_4$ signature of the gas produced in the WWTP is $-51.3 \pm 0.2$‰. 13 plume measurements next to the WWTP were taken over 5 days in October 2016 to February 2017. The $CH_4$ peak heights varied between 2.4 and 8.5 ppm. The isotopic signature for the seven used plume measurements are within the range of $-49.4$ to $-56.3$‰ with an average daily mean of



$-52.5\pm1.4‰$. This agrees well with the results of Zazzeri (2016) who reported isotopic signatures of $CH_4$ between $-48.1$ to $-59.2‰$ for wastewater treatment emissions.

### 3.2.5   Natural gas facilities

Besides the direct sampling of natural gas in Heidelberg (see Sect. 3.1) the plumes at two natural gas facilities were measured to determine the isotopic signature of $CH_4$ from natural gas in the region of Heidelberg. Between July 2016 and March 2017 the $CH_4$ concentration around the natural gas storage site in Sandhausen was measured over 10 days. Except for 2 days, the $CH_4$ concentration of the plumes was lower than 2.15 ppm and four times no significant changes of $CH_4$ could be measured at all. On these 2 days the maximal $CH_4$ concentration of the plume was higher with values between 2.3 and 10 ppm, so that the isotopic source signature could be determined with the Miller-Tans approach. The resulting $\delta^{13}CH_4$ signature was on average $-45.5\pm5.2‰$. $-41.8\pm0.4‰$ (two AirCores) on one day and $-49.2\pm4.6‰$ (one AirCore) on the other day. The natural gas storage in Sandhausen emitted only small amounts of $CH_4$ except during some events making it difficult to monitor.

Between Hähnlein and Gernsheim a natural gas storage, compressor stations and other natural gas facilities were placed together on one site. Over 5 days between September 2016 and February 2017 mobile measurements were carried out next to this site and showed that natural gas escaped at different locations. Contrary to the natural gas storage in Sandhausen, the measured $CH_4$ plumes had always maximum $CH_4$ concentrations mostly between 2.2 and 6 ppm, but some plumes even reached 6 to 25 ppm. Therefore, emissions from natural gas facilities are not negligible and seem to be highly heterogeneous. The determined $\delta^{13}CH_4$ signature of the $CH_4$ plumes was between $-41.1$ and $-57.4‰$. The average daily mean was $-46.6\pm6.8‰$ and thus a little bit less enriched than the isotopic signatures of $CH_4$ measured in Sandhausen and than the natural gas samples taken in Heidelberg ($-43.1\pm0.8‰$). The location of this natural gas facility can be the explanation for more depleted values, because the gas pipeline MEGAL passes this site directly and has a compressor station there. MEGAL runs from the border of the Czech Republic to France and mainly transports Russian natural gas, which has a more depleted isotopic signature.

### 3.2.6   Coal mines

On 25 March 2017 the emitted $CH_4$ concentration from bituminous deep coal mines in Bottrop were measured. In particular, the plume of one closed mine shaft and two ones that are still in service were measured. In the plume of the closed mine shaft the maximum $CH_4$ concentration measured was between 2.2 and 2.6 ppm while for the open mine shafts concentrations between 3 and 7.5 ppm were detected, although the mobile measurements were carried out much closer to the closed mine shaft. It seems that the $CH_4$ emissions from open mine shafts are larger than from closed ones. $\delta^{13}CH_4$ of the closed mine shaft is $-50.0\pm6.3‰$ while for the active mine shafts the average $\delta^{13}CH_4$ signature is $-56.0\pm2.3‰$ ($-54.7$ to $-59.5‰$). However, only one AirCore was measured for the closed mine shaft and the error of the isotopic source signature is larger than our criterion of 5‰. The determined isotope ratios of $CH_4$ in Bottrop match the coal bed gas samples from Bottrop ($-47.1$ to $-52.4‰$) measured by Thielemann et al. (2004). In addition, the values are similar to the average isotope ratio of $CH_4$ of $-55‰$ measured for $CH_4$ from bituminous coal in deep mines by Zazzeri et al. (2016).





On 23 March 2017 mobile measurements of $CH_4$ were carried out in the area around the lignite opencast mines Hambach and Garzweiler. However, no $CH_4$ emitted by the opencast mines directly could be detected. On the roads where the measurements were performed, the emission plume of $CH_4$ from these mines is apparently below the detection limit of the mobile system. High $CH_4$ concentrations were only measured at two locations next to the opencast pit. However, the two detected $CH_4$ plumes were measured upwind of the opencast mine and thus did not originate from the pit itself, but from the drainage system. The measured peak heights of the plumes were between 3 and 7.5 ppm. The $\delta^{13}CH_4c$ signature of the measured $CH_4$ is between $-79.7$ and $-84.8‰$ with an average of $-82.0 \pm 2.6‰$. These extremely depleted values indicate that the measured $CH_4$ is of microbial origin and thus is probably produced by $CO_2$-reduction similarly to one gas sample measured by Thielemann et al. (2004) with values of $-85.1$ to $-85.9‰$.

## 4    Conclusions

We have developed and tested a mobile instrument setup to determine the $\delta^{13}CH_4$ signature by measuring the plume of different $CH_4$ sources. The advantage of such a mobile application is that measurements can be performed downwind of the emission source and therefore outside of any industrial installation such as a gas compressor station or landfill without the consent of the owners. For accurate results, a carefully characterisation of the analyser, especially the cross sensitivities of $C_2H_6$, and the drying of air previous to the measurement is required. To reduce the $H_2O$ concentration below 0.1 % a nafion dryer was installed in the mobile setup and the cross sensitivity between $C_2H_6$ and the measurement of $\delta^{13}CH_4$ was corrected as shown in Fig. 3. Especially for natural gas samples, the precise determination and correction of $C_2H_6$ is important as in our study $C_2H_6$ can bias $\delta^{13}CH_4$ up to 3‰ depending on the $CH_4$ to $C_2H_6$ ratio of the sample and the calibration cylinder.

For the precise determination of the isotopic signature of different $CH_4$ sources we suggest to use the Miller-Tans approach using a York fit for most accurate results. There are three major limitations to the precise determination of the $\delta^{13}CH_4$ source signature: the number of data points during plume crossing, the measured concentration enhancement and the precision of the analyser for isotope analysis. The amount of data points limits the accuracy as the uncertainty decreases with increasing number of data points. To enlarge the amount of points the measurement should be carried out while driving as slowly as possible through the plume and then the plume should be remeasured using the AirCore. It is important to use the AirCore because it is a simple option to reduce the uncertainty by more than half. The most important limitation of the $\delta^{13}CH_4$ source signature is the plume concentration above background. Measured plumes with a peak height above background smaller than 0.45 ppm have uncertainties larger than 5‰ and thus are not used in this study. Driving as close as possible to the source increases the $CH_4$ concentration. However, where it is not possible, or the increase is not enough, the isotopic signature of the source cannot be determined with a sufficient precision with this method. To get better results even for smaller enhancements, more precise instruments are required in the future.

In this study, the $\delta^{13}CH_4$ signature of $CH_4$ emitted from a biogas plant, a landfill, dairy farms, a wastewater treatment plant, natural gas storage and compressor stations and bituminous deep mines were determined. The $\delta^{13}CH_4$ signatures measured during mobile campaigns are in good agreement with the measured isotope ratios from direct samples taken at some of the $CH_4$



sources and with values from other studies. Thus this method provides an opportunity to characterise the $CH_4$ emissions from a source where it is not possible or difficult to take direct samples; for example from an industrial site without the authorisation of the operating company, or from a large area where $CH_4$ emits heterogeneously at multiple unknown positions. Gas samples from Heidelberg city gas supply from December 2016 to June 2018 confirm a change in the natural gas mixture, especially

5    of Russian and North Sea gas. While in former years (1991 to 1996) strong seasonal variations of $\delta^{13}CH_4$ were measured, whereas recently the isotopic signature is nearly constant during the year. In addition, the average is approximately 2.8‰ more depleted than in the 1990s.

*Data availability.*   Data collected are presently available upon request.

*Competing interests.*   The authors declare that they have no conflict of interest.

10   *Acknowledgements.*   The authors thank to Michael Sabasch and Ingeborg Levin for calibration of standard gases and to Simone Wald for testing natural gas sample measurement. We thank Mr. and Mrs. Pfisterer from biogas plant Pfistererhof, Mr. Erdmann, Mr. Schröder and Mr. Schulz from Milchhof Weinheim for their help during our visits. Special thanks to Dr. Schmithausen from the University of Bonn, who welcome us at Haus Riswick in Kleve, for the visit and helpful discussions. The support from Mr. Blumenthal from AVR Kommunal GMbH was appreciated. Evan Cooper Border and Ellis King are acknowledged for prove reading this article.

15     The CRDS analyser was funded through the DFG excellence initiative II. We are grateful to the MeMo2 project (Marie Sklodowska-Curie grant agreement No 722479), which supported the measurement campaign to North Rhine-Westphalia.



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




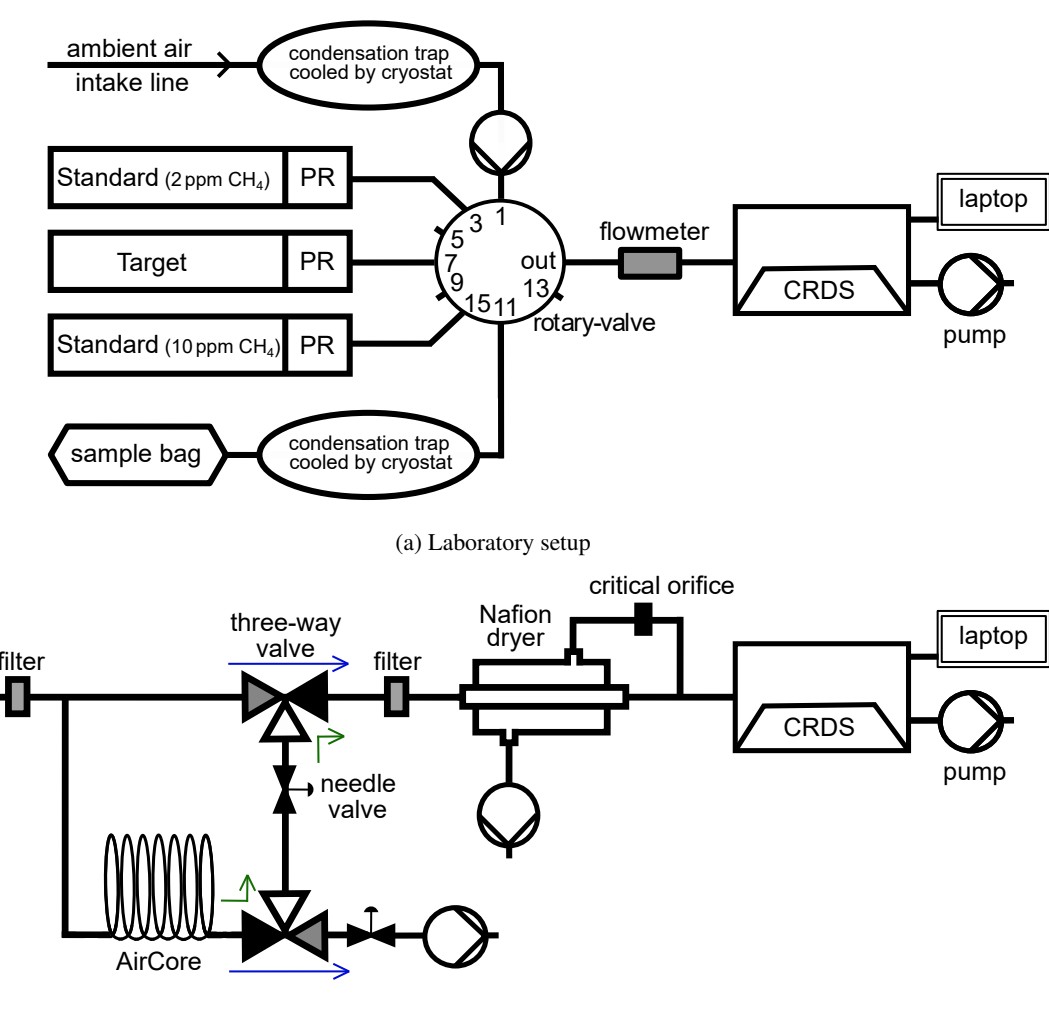

(a) Laboratory setup

(b) Mobile setup

**Figure 1.** Setup for measurements in the laboratory and with the mobile platform.

Figure (a) shows the measurement setup in the laboratory. Over port 1 ambient air measurements are performed. Port 11 is used to measure sample bags. Standard gas and target cylinders are measured on port 3, 7 and 15 to calibrate the above mentioned measurements and also the mobile ones.

Figure (b) shows the mobile measuring setup installed inside a van. The blue arrows indicate the flow of air in 'monitoring mode' and the green ones in 'replay mode'.



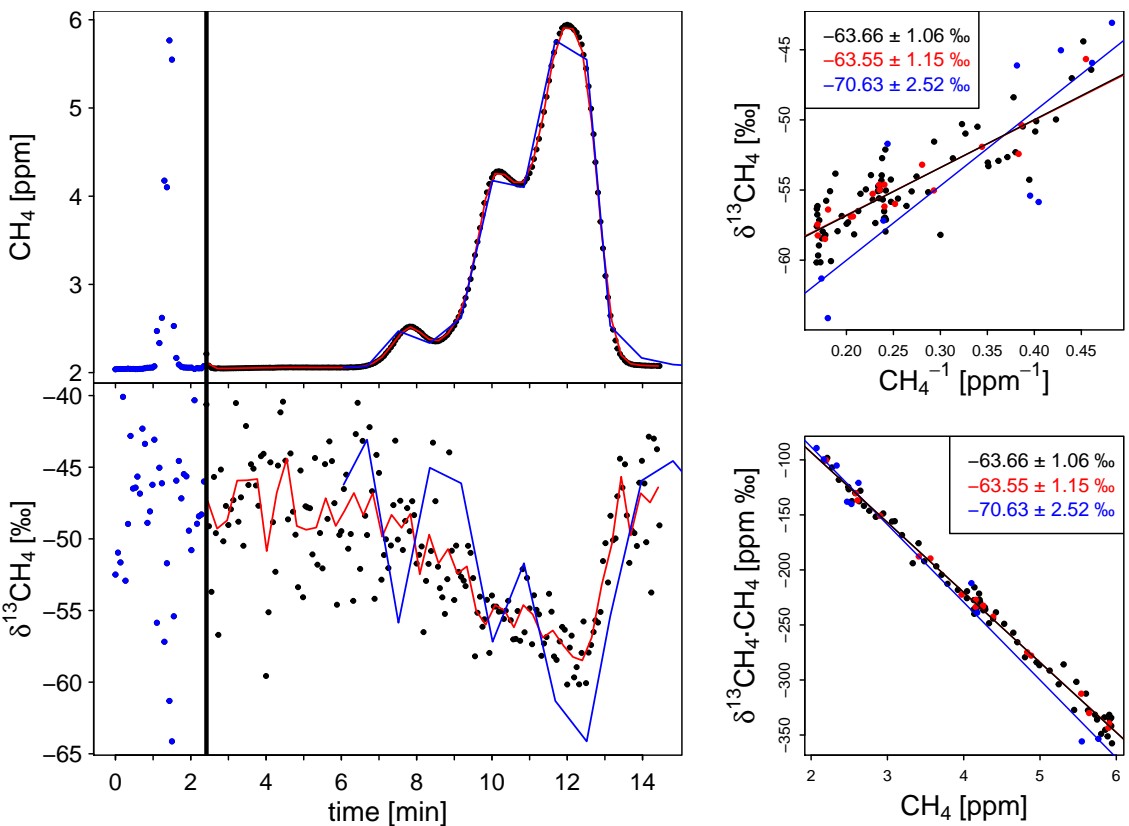

**Figure 2.** Measurement of a typical plume passing a biogas plant as well as Keeling Plot and Miller-Tans Plot to calculate the $\delta^{13}CH_4$ signature of the biogas plant.

Left: Typical $CH_4$ and $\delta^{13}CH_4$ peaks in the biogas plant plume. The vertical black line shows the switch from 'monitoring' to 'replay mode'. The red data are 15 sec average and the blue line is the in situ peak (first peak) stretched by a factor of 12.5.

Right: Keeling Plot (upper panel) and Miller-Tans Plot (lower panel) to calculate the $\delta^{13}CH_4$ source signature (insets). The blue colour represents the 'monitoring mode', the black and red (15 sec mean) ones the 'replay mode'. For better visibility the errorbars are not displayed.





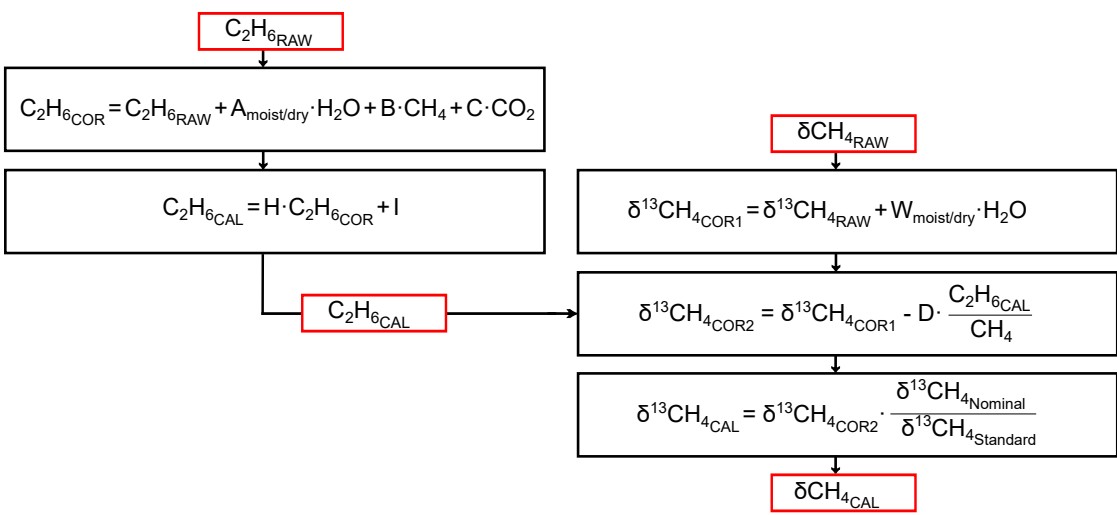

**Figure 3.** Scheme to correct and calibrate $C_2H_6$ and $\delta^{13}CH_4$. $\delta^{13}CH_{4_{Nominal}}$ is the nominal and $\delta^{13}CH_{4_{Standard}}$ the measured value of the calibration standard.





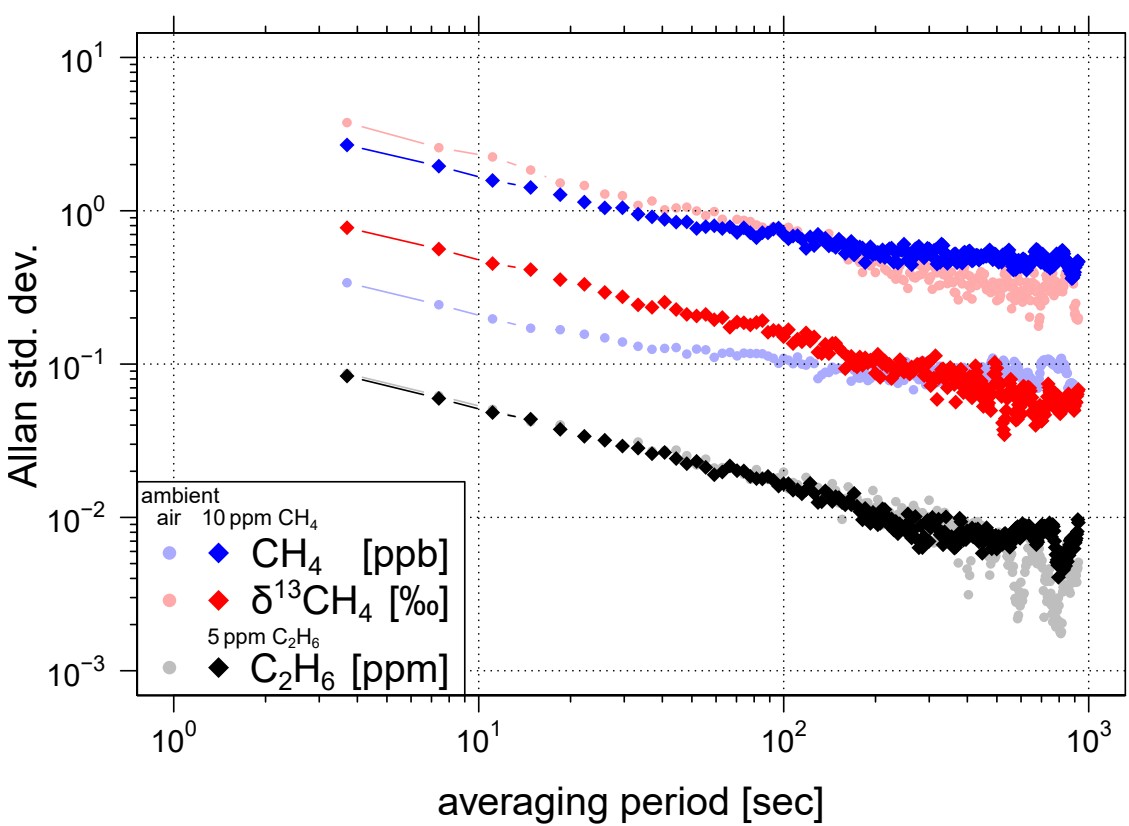

**Figure 4.** Allan standard deviations of $CH_4$, $\delta^{13}CH_4$ and $C_2H_6$ for two different sample gases each. Results for the first sample gas at atmospheric concentrations are shown in light (red, blue and grey) colours. Results for the second sample gas with 10 ppm $CH_4$ are shown in bright red and blue and for the third sample gas with 5 ppm $C_2H_6$ in black.





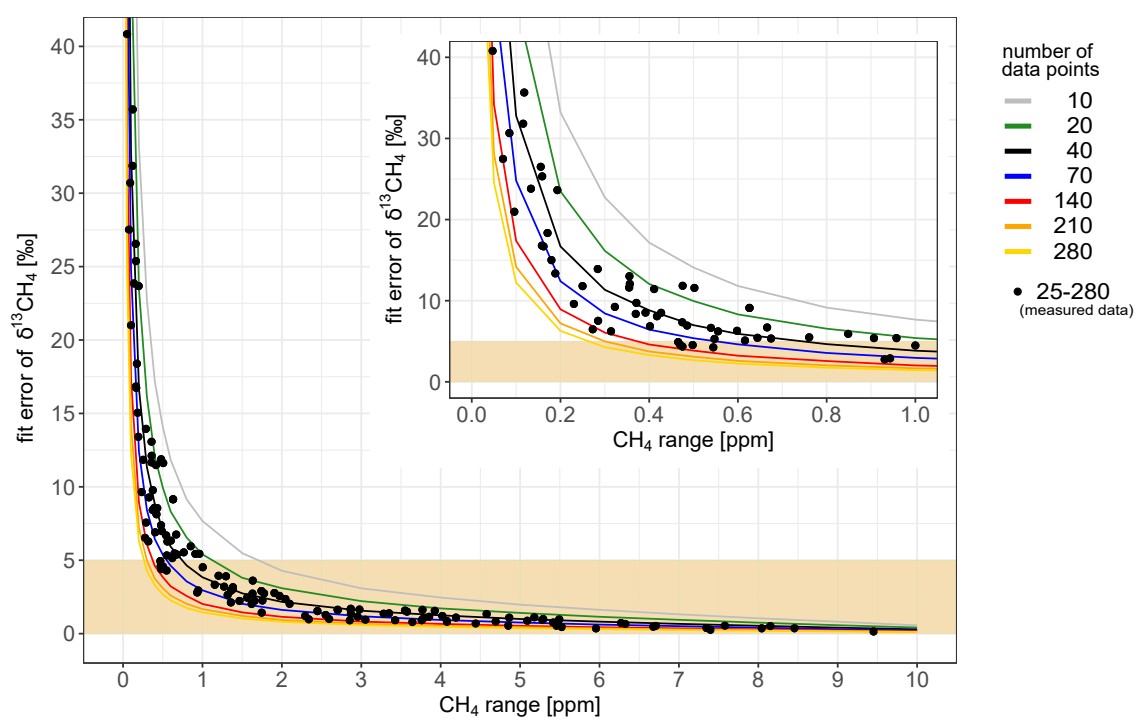

**Figure 5.** Dependency between peak height above background and error of the $\delta^{13}CH_4$ signature from the according measured peaks. The inserted figure shows an enlarged section with $CH_4$ ranges up to 1 ppm. The measured $\delta^{13}CH_4$ signatures with errors below 5‰ (data points within yellow shaded area) are used in this study. The lines show simulated data with different numbers of data points used in the Miller-Tans plot.



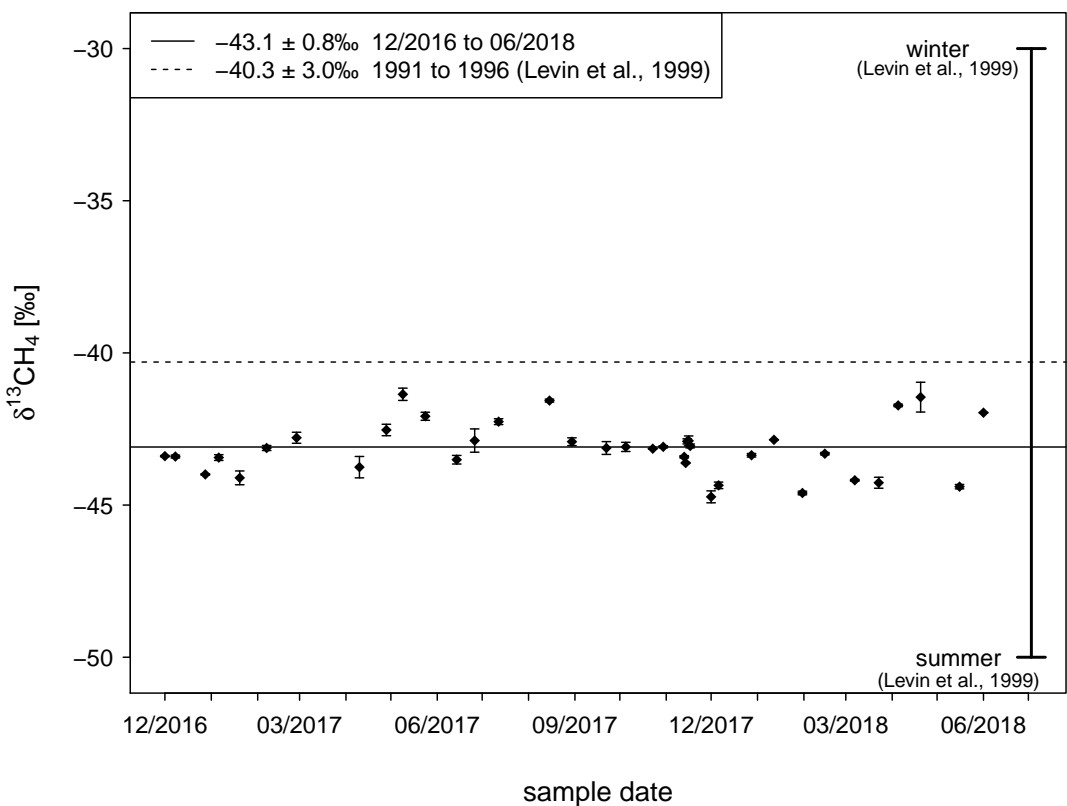

**Figure 6.** Isotopic signature of natural gas in Heidelberg measured between end of 2016 and June 2018. The horizontal solid line is the average monthly mean $\delta^{13}CH_4$ value. The horizontal dashed line is the average $\delta^{13}CH_4$ value measured from 1991 to 1996 with data ranging from $-50‰$ in summer to $-30‰$ in winter (Levin et al., 1999).




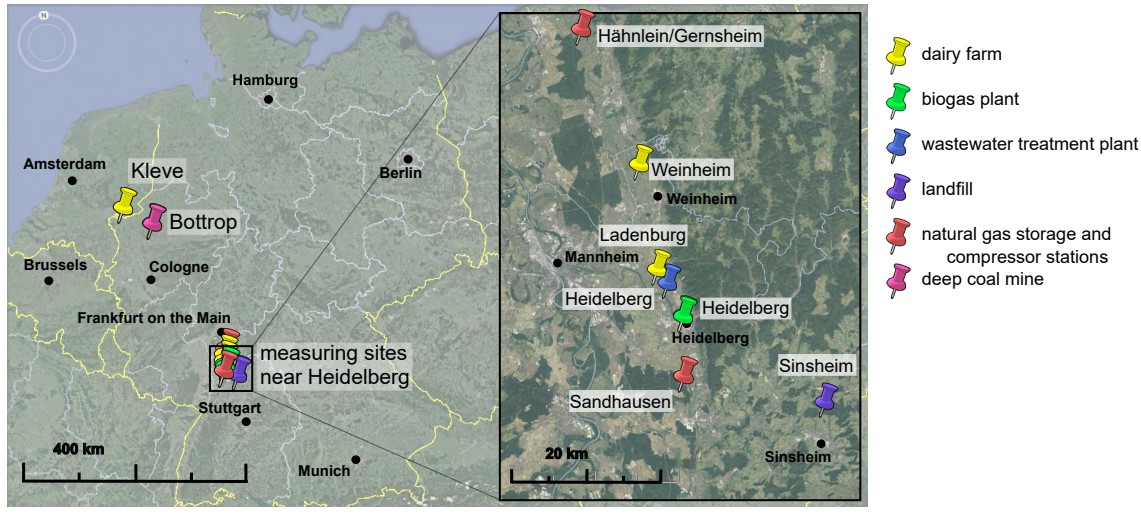

**Figure 7.** Locations of the measuring sites.




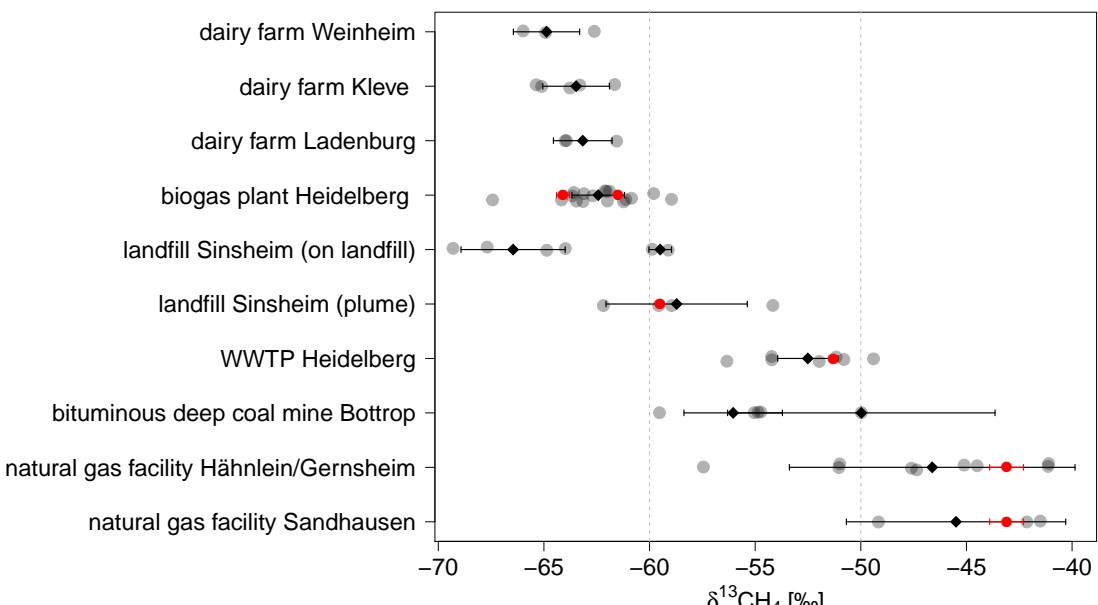

**Figure 8.** $\delta^{13}CH_4$ signature of $CH_4$ sources. Each determined $\delta^{13}CH_4$ signature is shown as grey dot. The black diamond shaped points show the averaged daily mean $\delta^{13}CH_4$ signature with the standard deviation. The $\delta^{13}CH_4$ values measured from gas samples taken at the different sites are plotted as red points. For both natural gas facilities it was not possible to take direct samples. Here the red points indicate the mean $\delta^{13}CH_4$ signature of natural gas in Heidelberg measured between end of 2016 and June 2018 as described in this study.



**Table 1.** Correction and calibration factors for $C_2H_6$ and $\delta^{13}CH_4$.

| influence of | | correction/ calibration factor | unit | method | tested range |
|---|---|---|---|---|---|
| $H_2O$ on $\delta^{13}CH_4$ | $W_{moist}$ | $-0.54 \pm 0.29$ | $(‰\, \delta^{13}CH_4)\,(\%\,H_2O)^{-1}$ | humidity tests | 0.16 to 1.5 % $H_2O$ |
| | $W_{dry}$ | $-$ | | | up to 0.15 % $H_2O$ |
| $H_2O$ on $C_2H_6$ | $A_{moist}$ | $0.70 \pm 0.10$ | $(ppm\, C_2H_6)\,(\%\,H_2O)^{-1}$ | humidity tests | 0.16 to 1.5 % $H_2O$ |
| | $A_{dry}$ | $0.43 \pm 0.51$ | $(ppm\, C_2H_6)\,(\%\,H_2O)^{-1}$ | | up to 0.15 % $H_2O$ |
| $CH_4$ on $C_2H_6$ | B | $0.0077 \pm 0.0007$ | $(ppm\, C_2H_6)\,(ppm\, CH_4)^{-1}$ | dilution & injection tests | 2 to 10 ppm $CH_4$ 0 to 1 ppm $C_2H_6$ |
| $CO_2$ on $C_2H_6$ | C | $(1.25 \pm 0.94)\cdot10^{-4}$ | $(ppm\, C_2H_6)\,(ppm\, CO_2)^{-1}$ | dilution & injection tests | 10 to 600 ppm $CO_2$ 0 to 1.3 ppm $C_2H_6$ |
| $C_2H_6$ calibration | H | $0.538 \pm 0.002$ | | dilution tests | 0 to 3 ppm $C_2H_6$ |
| | I | $0.070 \pm 0.005$ | ppm | | |
| $C_2H_6$ on $\delta^{13}CH_4$ | D | $40.87 \pm 0.49$ | $(‰\, \delta^{13}CH_4)\left(\frac{ppm\, C_2H_6}{ppm\, CH_4}\right)^{-1}$ | dilution tests | up to 0.7 $(ppm\, C_2H_6)(ppm\, CH_4)^{-1}$ |



**Table 2.** Determined $\delta^{13}CH_4$ signatures of $CH_4$ sources.

| location | $\delta^{13}CH_4$ signature from mobile measurements | | $\delta^{13}CH_4$ signature of direct gas samples | peak height* | number of AirCores** | number of visits** | mobile measuring period/dates |
| --- | --- | --- | --- | --- | --- | --- | --- |
| | average [‰] | range [‰] | [‰] | [ppm] | | | [MM,YY] |
| **biogas plant** | | | | | | | |
| Heidelberg | $-62.4 \pm 1.2$ | $-67.4$ to $-59.0$ | $-61.5 \pm 0.1$ $-64.1 \pm 0.3$ | 3.4 to 14.1 | 17 (25) | 7 (10) | Aug,16 to Mar,17 |
| **dairy farm** | | | | | | | |
| Weinheim (on farm) | $-64.9 \pm 1.6$ | $-66.0$ to $-62.6$ | | 8.3 to 8.9 | 3 (3) | 2 (2) | Oct,16 and Nov,16 |
| Weinheim (plume with biogas plant) | $-54.0 \pm 8.0$ | $-62.6$ to $-43.1$ | | 3.9 to 13.1 | 10 (12) | 5 (5) | Sep,16 to Feb,17 |
| Ladenburg (on farm) | $-63.2 \pm 1.4$ | $-64.0$ to $-61.6$ | | 4.1 to 7.3 | 3 (3) | 1 (1) | Oct,16 |
| Ladenburg (plume with biogas plant) | $-44.4 \pm 7.2$ | $-55.1$ to $-40.3$ | | 3.9 to 8.2 | 3 (8) | 1 (3) | Nov,16 to Feb,17 |
| Kleve | $-63.5 \pm 1.6$ | $-65.1$ to $-61.7$ | | 4.7 to 13.6 | 5 (5) | 1 (1) | Mar,17 |
| **landfill** | | | | | | | |
| Sinsheim (plume) | $-58.7 \pm 3.3$ | $-62.2$ to $-54.2$ | $-59.5 \pm 0.1$ | 2.4 to 2.6 | 4 (18) | 4 (8) | Jul,16 to Mar,17 |
| Sinsheim (on landfill) | $-59.5 \pm 0.5$ | $-59.9$ to $-59.1$ | | 3.9 to 7.2 | 2 (4) | 1 (1) | Jul,17 |
| | $-66.5 \pm 2.5$ | $-69.3$ to $-64.0$ | | 2.6 to 6.0 | 4 (4) | 1 (1) | Jul,16 |
| **WWTP** | | | | | | | |
| Heidelberg | $-52.5 \pm 1.4$ | $-56.3$ to $-49.4$ | $-51.3 \pm 0.2$ | 3.5 to 6.0 | 7 (13) | 5 (5) | Oct,16 to Feb,17 |
| **natural gas facilities** | | | | | | | |
| Sandhausen | $-45.5 \pm 5.2$ | $-49.2$ to $-41.5$ | | 3.0 and 10.0 | 3 (9) | 2 (10) | Jul,16 and Mar,17 |
| Hähnlein/Gernsheim | $-46.6 \pm 6.8$ | $-57.4$ to $-41.1$ | | 3.3 to 8.2 | 9 (21) | 5 (5) | Sep,16 to Feb,17 |
| **bituminous deep coal mine** | | | | | | | |
| Bottrop (active) | $-56.0 \pm 2.3$ | $-59.5$ to $-54.7$ | | 3.4 to 7.6 | 4 (4) | 1 (1) | Mar,17 |
| Bottrop (closed) | $-50.0 \pm 6.3$ | $-50.0$ | | 2.6 | 1 (1) | 1 (1) | Mar,17 |

\* The range of peak heights of the applied peaks measured with the AirCore.

\*\* Instead of the used AirCore measurements and the coresponding visits, the number in brackets refer to all AirCore measurements and visits.