# Peer review of "Improved method for mobile characterisation of $\delta^{13}$ CH4 source signatures and its application in Germany"

_Atmospheric Measurement Techniques, 2018_

## Referee Comment (RC1) · Anonymous Referee #1 · 30 Oct 2018

AMT-2018-259 - Review

General Comments:

This is a well written manuscript describing a very careful set of measurements of local isotopic carbon source signatures. The authors rightly point out that high quality measurements of source signatures is a critical aspect of using isotopes to quantify and attribute regional and global methane emissions. This paper, building on previous work, demonstrates practical methods and analytical guidance for quantifying isotopic source signatures. The authors are careful to report uncertainties at every stage of the process, which is important to all isotope analysis.

The writing is clear and concise, and the English usage and grammar is excellent.

Specific Comments:

One clear omission is the C2H6 / CH4 signatures from the individual plume measurements. A great deal of important work is done in calibrating the C2H6 channel from the instrument, but then the field data from this channel is not presented. Reporting these ratios would serve to validate that the instrument is reporting reasonable values; conversely, if the ratios do not make sense (e.g., if the landfill C2/C1 ratio is not zero within uncertainties), then that may point to an additional unknown systematic error in the system. For example, these data could be easily added to Table 2.

Similarly, for the bag samples of the local natural gas source, the measurements of the ethane content would be an interesting addition to the manuscript. Furthermore, the ethane content in the natural gas infrastructure is generally recorded by the natural gas distribution company; the values obtained from the analysis could be compared to the known values.

A second aspect of the manuscript that could use more attention is the calibration of d13C-CH4. Hoheisel (2017) is cited, but at least a brief description should be provided here (and/or a more detailed description in the supplemental material). For example, there are terms in Fig 3 (e.g., dCH4\_Nominal) which are not even described in the text. This is critical information if these data are to be used in partitioning of global or regional methane emissions. It is not clear from the text or the figure whether standards with a range of isotope ratios was used in the calibration; please clarify.

Detailed Comments and Questions:

- P1L3: "Therefore..." poor word usage. Rewrite.
- P3L17: polypropylene
- P3L27: 260 amp-hr. check units
P3L28: Supplier & Model Number for the inverter

P4L3: How were the valves triggered for "replay" mode? Add a description.

P6L24-25: Relative increase of which uncertainty? This statement is unclear. Please clarify.

P7L29: Please provide a brief description of the York fit.

P8L12: Why do the Keeling and MT methods produce identical source signatures and uncertainties (as shown in the figure)? I assume that this is a error in the preparation of the figure. If not, please explain.

P10L13: It's surprising to me that north sea gas is high in ethane content and heavy in isotope ratio. Generally heavy isotope ratios are associated with low C2+ content. Please confirm these values.

P11L33 (for example). The plume peak heights should be reported above the local baseline value, rather than absolute methane values. Please clarify in the description (e.g., "Peak height above baseline")

P12 Landfill section: The isotope signature of landfills are complicated by a) the isotope signature of the source material in the landfill, this fractionation that occurs by the anaerobic bacteria, and the further fractionation that can occur in the landfill cap where the methane is consumed aerobically.

P13: WWTP: The methane emissions from WWTPs can come from various locations and processes within the facility, and these processes may have different isotope signatures. Further, different WWTP plants employ different techniques to digest the waste; some discussion of these processes and their likely impact on your measurements would be helpful.

---

## Referee Comment (RC2) · Anonymous Referee #2 · 3 Nov 2018

This paper investigates a method for unbiased isotopic measurements of methane using a commercially available cavity ring-down spectrometer in a mobile setup. It specifically provides guidance on the correction for potential biases from ethane and water in the measured gases. These methods are then used to measure and characterize methane isotopic signatures from different sources in Germany.

Given the increasing interest in isotopic measurements of methane for source attribution studies and the availability of relatively low-cost analyzers, this paper is a useful addition to the literature to help researchers improve their measurements. The tables and figures are well presented. No major shortcomings were noticed. However, below

is a list of detailed comments that may help clarify arguments and language, and correct potential errors. Additionally, I suggest altering the title to something like "Improved method for mobile characterisation of  $\delta$ 13CH4 source signatures and its application in Germany." The current title sells short of a major contribution of the paper.

Detailed comments follow:

1. P 1, In 3: Remove "Therefore"; This sentence doesn't logically follow from the previous sentence. It is detail in addition to the previous sentence.

2. P 1, In 5: Explain gas matrix or replace with less jargon.

3. P 1, In 7: Abundant in many natural gases, but not all. Dry gas regions can contain only very small traces of ethane.

4. P 1, In 16: A 2.8 per mil difference begs the question whether the above mentioned up to 3 per mil ethane bias could have played a role here.

5. Switching between "CH4" and "methane" throughout the MS. Check for consistency.

6. P 1, In 20: Use original data references instead of reviews, e.g., https://agupubs.onlinelibrary.wiley.com/doi/full/10.1029/2009GL039780

7. P 2, In 1: You probably mean biomass burning.

8. P 2, In 2: More accurate to say "e.g. a sub-category of fossil fuel extraction". Some gases extracted by the fossil fuel industry is biogenic.

9. P 2, In 7: Do you mean to say that the isotopic signal was used to diagnose methane emission reductions? Not clear as written.

10. P 2, In 13: What do you mean with "all of these seasonal variations"?

11. P 2, In 26: Briefly describe the importance/use of a storage tube (first mention).

12. P 2, In 26: Nine facilities in 21 campaigns? Do you mean 21 measurement days?

AMTD
13. P 3, In 15: Perhaps that's explained later, but how do you get so close to the source that you're able to sample between 50 and 90% CH4?

14. P 3, In 17: Specify the composition of synthetic air.

15. P 3, In 33: Do you mean a 20-25 sec lag time between air sampling at inlet and fully arriving in the cavity? Sounds like it, but not fully clear.

16. P 4, In 11: What do you mean with representations have the same width? They don't. It's a bit confusing. The next sentence already states that both peaks represent the same emission plume.

17. P 4, In 13: Any hypotheses why this may be the case? This seems to be an important observation.

18. Table 1: Is the unit for ethane sensitivity to water (ppm) a typo? Units of ppb would make more sense. For reference, atmospheric ethane background concentrations are in the order of

24. P 10, In 13: What are the uncertainties for the ethane-to-methane ratios?

25. P 12, In 2: Why were the plumes expected to be smaller than on the other farm?

26. P 12, In 11: Why are seasonal differences for the biogas plant expected?

27. P 12, par. 1 & 3: Is C3/C4 diet information available for the Ladenburg and Kleve farms?

28. P 14, In 11: How small the fluxes were actually depends on the size of the facility (is it possible to detect all plumes at once?) and the wind conditions in addition to the measured methane concentrations.

29. P 14, In 20: Did your measurements include incomplete combustion from the compressor turbines? This could be detected by the associated CO2 or CO signal. It is still an open question whether high heat leads to isotopic fractionation of the source.

30. P 14, In 25: Do you use "open" and "in service" interchangeably? Not clear. If yes, it's not a surprise that a mine currently in service produces more emissions than a closed mine.

31. P 14, In 30: The Bottrop mine shaft measurements do not match coal bed gas samples except for the closed mine with a large error bar and only one AirCore measurement.

32. Conclusions: In the first paragraph, it's important to highlight again that these results (including the ethane bias) are specific to the CRDS instrument used.

AMTD

---

## Referee Comment (RC3) · Anonymous Referee #3 · 6 Nov 2018

**General Comments**

This paper presents a measurement setup used to determine the isotopic signatures of methane sources in Germany. The measurements are done with a CRDS instrument allowing direct measurements of samples or mobile measurements close to the sources. The authors have used the AirCore method to precisely determine the CH4 isotopic signatures while crossing plumes of targeted sources.

The paper is well organized, and a particular attention has been addressed to present and correct the well-known cross-sensitivities introduced by the CRDS. The instrument is also well described and characterized, but the calibration strategy and the data quality insurance remain unclear to me. In addition, the part describing the mobile measurements and the AirCore technique lacks of details for a measurement technique paper (see my questions/comments in the specific comments).

The results are clearly described and compared with other studies. It is then much of interest for the scientific community as it participates to the general effort for increasing the scientific database of CH4 isotopic signatures. It is nevertheless regrettable that the C2C6 to CH4 ratios are not presented along with the isotopic signatures. I strongly suggest to add it in order to give the paper more visibility.

Specific comments

The language used in the manuscript is not always precise. For instance, the terms "13C values" and "13C signatures" are not the same and sometimes confusing as used in the MS. Also, the 13C signatures are not directly measured by the CRDS but determined after data treatment from the measurements. Please correct through the all manuscript.

Most of the time, the uncertainties are presented, but without any explanation on how they are calculated and which parameters are used to computed them. It is also not clear what is the difference between uncertainties and precision. For instance, page 9 line 5/6, it is stated that increasing the number of data point improves the uncertainties on signatures. The next sentence, "uncertainties" has been replaced by "precision". It is confusing. Please clarify through the all manuscript.

The G2201-i instrument can be used in different mode, which drives the measurement frequency of each species. Also, the methane is reported by the instrument in high range (HR) and high precision (HP) mode depending on the level of the measured mole fractions. This drives the instrumental precision. Please, add some details on these two points.

Part2.1.2 and figure1: here is a list of questions/comments which should be addressed:

- how are the valves switched? - how is the flow measured and/or controlled? -why the flow presented here differs so much from the laboratory setup (160ml/min vs 20 to 80ml/min)? - what is the typical air flow going through the AC in monitoring mode? - what is the typical vehicle speed while in monitoring mode? - in figure1, the blue and green arrows are difficult to differentiate. Please, make them thicker.

Part3: how do you make sure that the direct samples are not mixed with ambient air? This could bias the isotopic signatures.

Part 3.2: it has been shown in figure5 that the CH4 peak height above the background is mostly driving the fit error. It would make much more sense to present all the peak heights above the background instead of the absolute CH4 value.

Conclusion: the Miller-Tans and Keeling approaches give the same results. Why do you suggest using one instead of the other one?

P1, line 6 : C2H6 only affects the 13C measurements

P1, line 11: 13CH4 signatures instead of values

P2, line 1: "biogas burning" is not a CH4 source

P2, line 11: do you mean "due to its origin,"?

P2, line 15: introduce IRMS here and not line 17, and add what GC stands for.

P2, line 16: I suggest to delete "has been"  $\rightarrow$  as shown by Röckmann et al.

P3, line 1: replace signature by ratio. Signatures are not directly measured by the CRDS.

P3, line 13: it is surprising to observe such a flow range (factor 4). Could you explain why and give more details? Is the flow controlled somehow?

P3, line 17: what do you mean by synthetic air? Have you checked it is CH4 free?

P3, line 20: do you keep the 15min or is there a stabilization time?

**C3**

P3, line 25: decabon

P3, line 33: 160ml/min to be consistent with the previous part.

P4, line 5: How the measurement precision can be better in replay mode than in monitoring mode with the same instrument? Please clarify?

P4, line 30: have you tested the Nafion for potential fractionation?

P5, line 29: Assan et al. 2017 showed that the intercept changes in time due to baseline drift. Have you regularly checked it?

P5, line 33: please, check the unit

P6, line 1: what do you mean by fully?

P6, line 6: you cannot get a concentration range with a single cylinder. Please reformulate.

P6, line 8: as I understood, a one point calibration strategy is used, meaning that you assume that the instrument has a linear response through the all measurement scale (mobile and sample measurements). Then why two different cylinders are used as calibration gases?

P6, line 10: have you seen some changes in the CRDS regime before and after each experiments? What was the maximal drift observed and how are they taking into account?

P6, line 4 to 11: it is not clear to me how the CH4 is calibrated. There is no CH4 calibration factor in Table1.

P6, line 25: Please, detail how these uncertainties are calculated.

P6, line 32: is it the last 10min over the 15min measurements?

P7, line 5: is it a linear drift? The uncertainties is larger than the drift itself.

P7, line 19: please clarify which uncertainties you are talking about.

P7, line 23/24: were you driving while analyzing the AC? Micro-vibrations due to vehicle motion degrade the CRDS performances.

P7, line 25: do you mean uncertainties? Is that calculated only from the fits?

P8, line 17: isotopic signatures are not directly measured

P8, line 33: do you mean the fit error? Or is there more parameters used to derived the uncertainties?

P9, line 10: what precision?

P10, line 17/18 : these criteria are already described earlier, no need to state it again here.10  $\,$

P10, line 20: why is the daily mean used and not the single signatures? Same p13 line 6.

P11, line 30: what is the peak height of the third AC? Please replaced value with signature.

P14, line 10; replace the dot with and.

P14, line 11: I suggest to replace monitor by sample.

P14, line 15: choose between "always" and "mostly"

P14, line 16: only peak heights are measured, not fluxes. I would then delete "therefore" and add "from these natural gas facilities".

P15, line 3: what is the detection limit of the system? Are you sure there is plumes coming out?

P15, line 6: check for typo.

---

## Author Comment (AC1) · 21 Dec 2018

We would like to thank the reviewer for the helpful comments and suggestions. We answer each of them here after, with the original comment in blue and our response in black. We added the modifications done in the revised version in italics.

Specific Comments:

[Referee #1]   One clear omission is the $C_2H_6$ / $CH_4$ signatures from the individual plume measurements. A great deal of important work is done in calibrating the C2H6 channel from the instrument, but then the field data from this channel is not presented. Reporting these ratios would serve to validate that the instrument is reporting reasonable values; conversely, if the ratios do not make sense (e.g., if the landfill C2/C1 ratio is not zero within uncertainties), then that may point to an additional unknown systematic error in the system. For example, these data could be easily added to Table 2.
Similarly, for the bag samples of the local natural gas source, the measurements of the ethane content would be an interesting addition to the manuscript.
Furthermore, the ethane content in the natural gas infrastructure is generally recorded by the natural gas distribution company; the values obtained from the analysis could be compared to the known values.

[Hoheisel et al.]   Thank you for pointing this out, we have added an additional section (see below) and another figure.
*"$C_2H_6$ to $CH_4$ ratio of direct samples and mobile measurements*
*The $C_2H_6$ to $CH_4$ ratio of gas samples from the natural gas distribution network in Heidelberg varies between 0.027 and 0.072 with lower values in winter, 0.04±0.01, and higher ones in summer, 0.06 ± 0.01. This result can indicate that the percentage of Russian gas is higher in winter than in summer taking into account, that Nitschke-Kowsky et al., (2012) reported a $C_2H_6$ to $CH_4$ ratio for Russian natural gas to be 0.014 while for North Sea gas it is 0.078. Also the isotopic signatures of our natural gas samples support this trend with more depleted values in winter than in summer.*
*Gas emitted by other $CH_4$ sources like landfills, biogas plants and wastewater treatment plants do not contain $C_2H_6$. The $C_2H_6$ to $CH_4$ ratio of the gas samples taken directly at the gas collecting systems of these sources are zero within the errors and can be clearly separated from the natural gas samples (see Fig.7).*
*The separation due to $C_2H_6$ to $CH_4$ ratio works well with direct gas samples, but unfortunately not yet for mobile AirCore measurements. In contrast to the direct sample measurement, in mobile AirCore measurements $CH_4$ and $C_2H_6$ emitted by the source are diluted in the background. To determine the ratio, a linear fit of the measured $C_2H_6$ concentration to the measured $CH_4$ concentration is used. However, due to the high uncertainty of the $C_2H_6$ measurement without averaging as it is possible for the direct sample measurement, in combination with the very small or not existing changes in $C_2H_6$ the fit does not provide reasonable data."*

[Referee #1]   A second aspect of the manuscript that could use more attention is the calibration of d13C-CH4. Hoheisel (2017) is cited, but at least a brief description should be provided here (and/or a more detailed description in the supplemental material). For example, there are terms in Fig 3 (e.g., dCH4_Nominal) which are not even described in the text. This is critical information if these data are to be used in partitioning of global or regional methane emissions. It is not clear from the text or the figure whether standards with a range of isotope ratios was used in the calibration; please clarify.

[Hoheisel et al.]   We understand the point of the referee. For a better understanding, we deleted "and calibration" on page 4 line 22, since in this section we focus on corrections. In addition, we added some explanations in section 2.3.3 and in the caption of Figure 3. This section also provides more information about the used standard.
*"All data measured with the CRDS analyser in the laboratory or during mobile campaigns was corrected using the factors from Table 1 and following Fig.3 prior to the one point calibration calculation."*
*"The gas cylinder used for calibration was chosen according to the experiment to ensure a similar composition and similar concentrations for sample and standard."*
*"Scheme to correct and calibrate $C_2H_6$ and $\delta^{13}CH_4$. $\delta^{13}CH_{4\,Nominal}$ is the nominal (known value of the standard) and $\delta^{13}CH_{4\,Standard}$ the measured (or interpolated) value of the calibration standard."*

Detailed Comments and Questions:

[Referee #1]       P1L3: "Therefore..." poor word usage. Rewrite.

[Hoheisel et al.]    Agreed, we have changed it.

[Referee #1]       P3L17: polypropylene

[Hoheisel et al.]    Yes, corrected.

[Referee #1]       P3L27: 260 amp-hr. check units

[Hoheisel et al.]    Done.

[Referee #1]       P3L28: Supplier & Model Number for the inverter

[Hoheisel et al.]    Done, we now have included the Supplier and model number:
"*e-ast HighPowerSinus HPLS 1000-12*".

[Referee #1]       P4L3: How were the valves triggered for "replay" mode? Add a description.

[Hoheisel et al.]    We switch the valves manually after passing a methane plume. For clarification, we now have added the following phrase.
"*by switching the valves manually*".

[Referee #1]       P6L24-25: Relative increase of which uncertainty? This statement is unclear. Please clarify.

[Hoheisel et al.]    Ok, we have changed it.
"*the uncertainty of $\delta^{13}CH_4$*"

[Referee #1]       P7L29: Please provide a brief description of the York fit.

[Hoheisel et al.]    Thank you for pointing this out. We have now included a short description.
"*York's solution is the general LSE (least-squares estimation) solution: his equations provide the best possible, unbiased estimates of the true intercept a and slope b in all cases where the points are independent and the errors are normally distributed*" (Wehr and Saleska, 2017)

[Referee #1]       P8L12: Why do the Keeling and MT methods produce identical source signatures and uncertainties (as shown in the figure)? I assume that this is an error in the preparation of the figure. If not, please explain.

[Hoheisel et al.]    We appreciate the recommendation for clarity and we looked again into the matter. However, with both methods (KP and MT) the isotopic signature is determined identical within the first few decimal places when using the measured data and also for simulated datasets.

[Referee #1]       P10L13: It's surprising to me that north sea gas is high in ethane content and heavy in isotope ratio. Generally heavy isotope ratios are associated with low C2+ content. Please confirm these values.

[Hoheisel et al.]    We have checked it again and the values are correct.

[Referee #1]       P11L33 (for example). The plume peak heights should be reported above the local baseline value, rather than absolute methane values. Please clarify in the description (e.g., "Peak height above baseline")

[Hoheisel et al.]    We generally agree with the reviewer. However, in the text we wanted to give an expression of the $CH_4$ concentrations measured around the different sources. To clarify it, we change the phrase "peak height", when we report maximum $CH_4$ concentrations measured in the plume. Since it is correct, that the $CH_4$ peak heights above the background are mostly driving the fit error, we decided, to change the maximum values reported in Table 2 and in the supplements to peak heights above baseline.

[Referee #1]     P12 Landfill section: The isotope signature of landfills are complicated by a) the isotope signature of the source material in the landfill, this fractionation that occurs by the anaerobic bacteria, and the further fractionation that can occur in the landfill cap where the methane is consumed aerobically.

[Hoheisel et al.]     We now have added an additional sentence at the beginning of the section.
*"In addition to the source material and fractionation during the production the isotopic signature of gas emitted by landfills depends also on fractionation processes in the upper soil layers of the landfill. Due to the presence of oxygen in the upper soil layers, aerobic bacteria oxidate parts of $CH_4$ which diffuses through the soil cover and shift the isotopic composition to more depleted values. Bergamaschi et al. (1998) measured these different isotopic signatures."*

[Referee #1]     P13: WWTP: The methane emissions from WWTPs can come from various locations and processes within the facility, and these processes may have different isotope signatures. Further, different WWTP plants employ different techniques to digest the waste; some discussion of these processes and their likely impact on your measurements would be helpful.

[Hoheisel et al.]     We understand the concern of the referee and added a few minor explanations (see below). However, more details about the specific processes lead too far for this rather technical paper. Indeed, the study will be continued and another paper is planned that will describe the different processes in more detail.
*"There, the sludge treatment inside the digestion towers takes place in three septic tanks with a volume of $2500m^3$ each under anaerobic mesophilic conditions, that means without oxygen at $37°C$."*

[Figure]

**Figure 7.** Isotopic signature and $C_2H_6$ to $CH_4$ ratio of gas samples from a biogas plant, a landfill, a WWTP and the natural gas distribution system in Heidelberg measured between end of 2016 and November 2018.

---

## Author Comment (AC2) · 21 Dec 2018

We would like to thank the reviewer for the helpful comments and suggestions. We answer each of them here after, with the original comment in blue and our response in black. We added the modifications done in the revised version in italics.

[Referee #2] I suggest altering the title to something like "Improved method for mobile characterisation of 13CH4 source signatures and its application in Germany." The current title sells short of a major contribution of the paper.

[Hoheisel et al.] Thank you for pointing this out. We have changed the title.

Detailed comments follow:

[Referee #2] 1. P 1, ln 3: Remove "Therefore"; This sentence doesn't logically follow from the previous sentence. It is detail in addition to the previous sentence.

[Hoheisel et al.] Yes, corrected.

[Referee #2] 2. P 1, ln 5: Explain gas matrix or replace with less jargon.

[Hoheisel et al.] Ok, we have changed it.
*"To achieve precise results a CRDS analyser,…, was characterised especially with regard to cross sensitivities of composition differences of the gas matrix in air samples or calibration tanks".*

[Referee #2] 3. P 1, ln 7: Abundant in many natural gases, but not all. Dry gas regions can contain only very small traces of ethane.

[Hoheisel et al.] Yes, we have changed it.
*"$C_2H_6$ is typically abundant in natural gases".*

[Referee #2] 4. P 1, ln 16: A 2.8 per mil difference begs the question whether the above mentioned up to 3 per mil ethane bias could have played a role here.

[Hoheisel et al.] In the 1990s Levin et al. (1999) measured $\delta^{13}CH_4$ with a mass spectrometer after separation of $CH_4$ from other components of air and so there is no cross-sensitivity with $C_2H_6$.

[Referee #2] 5. Switching between "CH4" and "methane" throughout the MS. Check for consistency.

[Hoheisel et al.] Thanks, "methane" has been replaced in the manuscript by "$CH_4$".

[Referee #2] 6. P 1, ln 20: Use original data references instead of reviews, e.g., https://agupubs.onlinelibrary.wiley.com/doi/full/10.1029/2009GL039780

[Hoheisel et al.] Yes, corrected.

[Referee #2] 7. P 2, ln 1: You probably mean biomass burning.

[Hoheisel et al.] Thanks, corrected.

[Referee #2] 8. P 2, ln 2: More accurate to say "e.g. a sub-category of fossil fuel extraction". Some gases extracted by the fossil fuel industry is biogenic.

[Hoheisel et al.] Clarified.

[Referee #2] 9. P 2, ln 7: Do you mean to say that the isotopic signal was used to diagnose methane emission reductions? Not clear as written.

[Hoheisel et al.] Agreed, we have clarified it.

[Referee #2] 10. P 2, ln 13: What do you mean with "all of these seasonal variations"?

[Hoheisel et al.] Yes, this was unclear. We wanted to write "*seasonal variations*".

[Referee #2] 11. P 2, ln 26: Briefly describe the importance/use of a storage tube (first mention).

[Hoheisel et al.] Done.

[Referee #2] 12. P 2, ln 26: Nine facilities in 21 campaigns? Do you mean 21 measurement days?

[Hoheisel et al.] Yes, we have changed it to "*21 mobile measurement days*".

| | |
|---|---|
| [Referee #2] | 13. P 3, ln 15: Perhaps that's explained later, but how do you get so close to the source that you're able to sample between 50 and 90% CH4? |
| [Hoheisel et al.] | We take gas samples for example directly from the gas collecting system of the landfill and the WWTP. We now have clarified it and changed the phrase. "*Gas samples taken directly from different installations (e.g. natural gas pipelines, biogas plants, gas collecting systems of landfills and wastewater treatment plants) need to be diluted before the measurement with the CRDS analyser, because such samples usually consist of between 50 and 90% CH$_4$.*" |
| [Referee #2] | 14. P 3, ln 17: Specify the composition of synthetic air. |
| [Hoheisel et al.] | We now have added the composition (*20.5±0.5% O$_2$ in N$_2$*). |
| [Referee #2] | 15. P 3, ln 33: Do you mean a 20-25 sec lag time between air sampling at inlet and fully arriving in the cavity? Sounds like it, but not fully clear. |
| [Hoheisel et al.] | Yes, we clarified it. "*Due to the length of the intake line, the volume of the cavity, and a flow rate of 160ml/min the time lag between air sampling at inlet and measurement in the cavity of the CRDS analyser is 20 to 25sec.*" |
| [Referee #2] | 16. P 4, ln 11: What do you mean with representations have the same width? They don't. It's a bit confusing. The next sentence already states that both peaks represent the same emission plume. |
| [Hoheisel et al.] | What we meant is that when measuring a CH$_4$ plume first without and subsequently with the AirCore the two measured CH$_4$ peaks have the same height, but since we measured with a slower flow in monitoring mode not the same width. For better comparison we stretched the peak measured without the AirCore in x direction, to make it easier to compare the measurements done with and without the AirCore and to note that both peaks have the same shape and height. We also changed the text to: "*For comparison the peak measured in `monitoring mode' (blue dots/line on the left side) is stretched by a factor of 12.5 in x-direction (blue line on the right side) so that the peak measured with the AirCore and the stretched one measured without it have the same width. The peak measured in `replay mode' precisely corresponds to the stretched one measured in `monitoring mode', because both peaks reproduce the same emission plume.*" |
| [Referee #2] | 17. P 4, ln 13: Any hypotheses why this may be the case? This seems to be an important observation. |
| [Hoheisel et al.] | As described above we measured the air stored in the AirCore directly after we measured a CH$_4$ peak in monitoring mode. So the storage time is relatively short and we expect that the peaks measured with and without the AirCore have the same shape and the same height. |
| [Referee #2] | 18. Table 1: Is the unit for ethane sensitivity to water (ppm) a typo? Units of ppb would make more sense. For reference, atmospheric ethane background concentrations are in the order of < 1 ppb. |
| [Hoheisel et al.] | No, the unit ppm is correct. |
| [Referee #2] | 19. P 5, ln 5: Natural gases on a continuum between 0 to 40% ethane have been measured. See Sherwood et al., 2017 (already in your refs). Hence, the importance of ethane correction varies: very important for wet gas basins (meaning lots of associated gas) and less important for very dry gas regions (mature thermogenic dry gas). |
| [Hoheisel et al.] | Thank you for pointing this out. In this study we measured natural gas samples from the natural gas network in Heidelberg. Therefore, we wanted to give an overview of C$_2$H$_6$ contents of natural gases in the pipeline network in Germany. We have now added: "*As typical natural gases in the pipeline network in Germany contain…*" |

| [Referee #2] | 20. P 5, ln 10: where does the 3 per mil value come from? |
|---|---|
| [Hoheisel et al.] | The value of 3‰, is the bias $\delta^{13}CH_4$ has in presence of a $C_2H_6$ to $CH_4$ ration of 0.073, which was the highest value we measured for our natural gas samples. To make the text more understandable, we changed it. |

*"A correction is necessary because for typical $C_2H_6$ to $CH_4$ ratios between 0.027 and 0.073 measured for our natural gas samples, $\delta^{13}CH_4$ shows a bias between 1 and 3‰ to more enriched values. We must also keep in mind that similar shifts in, $\delta^{13}CH_4$ to less enriched values can occur when using a calibration cylinder which contains $C_2H_6$."*

| [Referee #2] | 21. P 7, ln 3: What is the sign of the drift? Is the drift due to fractionation in the bag or due to leakage of background air into the bag? |
|---|---|
| [Hoheisel et al.] | The sign is + and the major cause for the drift is fractionation by the leakage of sample air out of the bag. So for a better understanding we added the phrase "*to more enriched values"* and the following sentence. |

*"The drift occurs especially due to fractionation by diffusion of air through the sample bag".*

| [Referee #2] | 22. P 7, ln 28: What are simulated data? |
|---|---|
| [Hoheisel et al.] | How we simulated data is explained a few lines below. For better understanding we changes this section: |

*"Similar to the method described by Wehr and Saleska (2017) for $CO_2$ and $\delta^{13}CO_2$, we create several typical emission plume crossings. We generated synthetic $CH_4$ peaks using a background concentration of 1.95 ppm $CH_4$ and a Gaussian curve with 10-280 equidistant data points every 3.7 s and an enhancement of 100-10000 ppb. The corresponding $\delta^{13}CH_4$ values were calculated with $CH_4$ source signatures between -35‰ and -65‰ and a background of -48‰. To reproduce the statistical uncertainties of a real measurement, we add a normally distributed scattering around zero to the synthetic $CH_4$ concentrations and the corresponding isotope ratios. The standard deviation of the normal distributed scattering depends on the $CH_4$ concentrations and was chosen as the Allan standard deviation measured for raw data of the analyser. However, when simulating possible improved analysers, we reduced the scattering by a factor 2 to 10.*
*Such sets of data were generated 5000 times for each condition. To study the influence of the averaging time, we calculate mean data sets with varying averaging periods (up to 1min).*
*For each dataset the $\delta^{13}CH_4$ source signature was calculated with the Miller-Tans and the Keeling method using the York or the OLS fit."*

| [Referee #2] | 23. P 10, ln 4: I'm confused. Here it says no significant seasonal cycle has been observed. In the next paragraph, it says the values are more depleted in winter than in summer. Are there enough samples to determine this correctly? |
|---|---|
| [Hoheisel et al.] | We changed this paragraph. |

| [Referee #2] | 24. P 10, ln 13: What are the uncertainties for the ethane-to-methane ratios? |
|---|---|
| [Hoheisel et al.] | The uncertainties for the ethane-to-methane ratios is in both cases 0.01. We now have included it. |

| [Referee #2] | 25. P 12, ln 2: Why were the plumes expected to be smaller than on the other farm? |
|---|---|
| [Hoheisel et al.] | Both farms have a different number of dairy cows. We now added the comment "*due to lower animal number".* |

| [Referee #2] | 26. P 12, ln 11: Why are seasonal differences for the biogas plant expected? |
|---|---|
| [Hoheisel et al.] | Thanks, we have removed "seasonal". |

| [Referee #2] | 27. P 12, par. 1 & 3: Is C3/C4 diet information available for the Ladenburg and Kleve farms? |
|---|---|
| [Hoheisel et al.] | No, unfortunately not. |

| | |
|---|---|
| [Referee #2] | 28. P 14, ln 11: How small the fluxes were actually depends on the size of the facility (is it possible to detect all plumes at once?) and the wind conditions in addition to the measured methane concentrations. |
| [Hoheisel et al.] | The facility is in a small forest and so in difficult terrain. It is unlikely that we detect all plumes at once. So we weakened our statement. |
| | |
| [Referee #2] | 29. P 14, ln 20: Did your measurements include incomplete combustion from the compressor turbines? This could be detected by the associated CO2 or CO signal. It is still an open question whether high heat leads to isotopic fractionation of the source. |
| [Hoheisel et al.] | To determine the origin of a $CH_4$ plume measured around the natural gas facility between Hähnlein und Gernsheim is difficult, because this facility contains a natural gas storage and several compressor stations even operated by different gas providers. At this site further work and more measuring campaigns are planned to receive more detailed results. |
| | |
| [Referee #2] | 30. P 14, ln 25: Do you use "open" and "in service" interchangeably? Not clear. If yes, it's not a surprise that a mine currently in service produces more emissions than a closed mine. |
| [Hoheisel et al.] | Yes, we changed "open" to "in service". |
| | |
| [Referee #2] | 31. P 14, ln 30: The Bottrop mine shaft measurements do not match coal bed gas samples except for the closed mine with a large error bar and only one AirCore measurement. |
| [Hoheisel et al.] | Done. |
| | |
| [Referee #2] | 32. Conclusions: In the first paragraph, it's important to highlight again that these results (including the ethane bias) are specific to the CRDS instrument used. |
| [Hoheisel et al.] | OK, we have added the following phrase to highlight your recommendation. "*characterisation of each individual analyser*" |

---

## Author Comment (AC3) · 21 Dec 2018

We would like to thank the reviewer for the helpful comments and suggestions. We answer each of them here after, with the original comment in blue and our response in black. We added the modifications done in the revised version in italics.

**General Comments**

[Referee #3] It is nevertheless regrettable that the C2C6 to CH4 ratios are not presented along with the isotopic signatures. I strongly suggest to add it in order to give the paper more visibility.

[Hoheisel et al.] We appreciate the recommendation and we added the short section $C_2H_6$ to $CH_4$ ratio of direct samples and mobile measurements as well as one Figure.

*"$C_2H_6$ to $CH_4$ ratio of direct samples and mobile measurements*
*The $C_2H_6$ to $CH_4$ ratio of gas samples from the natural gas distribution network in Heidelberg varies between 0.027 and 0.072 with lower values in winter, 0.04±0.01, and higher ones in summer, 0.06 ± 0.01. This result can indicate that the percentage of Russian gas is higher in winter than in summer taking into account, that Nitschke-Kowsky et al., (2012) reported a $C_2H_6$ to $CH_4$ ratio for Russian natural gas to be 0.014 while for North Sea gas it is 0.078. Also the isotopic signatures of our natural gas samples support this trend with more depleted values in winter than in summer.*
*Gas emitted by other $CH_4$ sources like landfills, biogas plants and wastewater treatment plants do not contain $C_2H_6$. The $C_2H_6$ to $CH_4$ ratio of the gas samples taken directly at the gas collecting systems of these sources are zero within the errors and can be clearly separated from the natural gas samples (see Fig.7).*
*The separation due to $C_2H_6$ to $CH_4$ ratio works well with direct gas samples, but unfortunately not yet for mobile AirCore measurements. In contrast to the direct sample measurement, in mobile AirCore measurements $CH_4$ and $C_2H_6$ emitted by the source are diluted in the background. To determine the ratio, a linear fit of the measured $C_2H_6$ concentration to the measured $CH_4$ concentration is used. However, due to the high uncertainty of the $C_2H_6$ measurement without averaging as it is possible for the direct sample measurement, in combination with the very small or not existing changes in $C_2H_6$ the fit does not provide reasonable data."*

**Specific comments**

[Referee #3] The language used in the manuscript is not always precise. For instance, the terms "13C values" and "13C signatures" are not the same and sometimes confusing as used in the MS. Also, the 13C signatures are not directly measured by the CRDS but determined after data treatment from the measurements. Please correct through the all manuscript.

[Hoheisel et al.] Yes, we clarified it.

[Referee #3] Most of the time, the uncertainties are presented, but without any explanation on how they are calculated and which parameters are used to computed them. It is also not clear what is the difference between uncertainties and precision. For instance, page 9 line 5/6, it is stated that increasing the number of data point improves the uncertainties on signatures. The next sentence, "uncertainties" has been replaced by "precision". It is confusing. Please clarify through the all manuscript.

[Hoheisel et al.] Yes, we corrected it.

[Referee #3] The G2201-i instrument can be used in different mode, which drives the measurement frequency of each species. Also, the methane is reported by the instrument in high range (HR) and high precision (HP) mode depending on the level of the measured mole fractions. This drives the instrumental precision. Please, add some details on these two points.

[Hoheisel et al.] We agree that this is an important point and added the following text.
*"All measurements with the CRDS analyser were done in the combined $CO_2/CH_4$ mode to measure $CH_4$ and $CO_2$ parallel. In addition, the High Precision (HP) mode for $CH_4$ is chosen to provide the most precise $CH_4$ measurements for $CH_4$ concentrations up to 12 ppm."*

| | |
|---|---|
| [Referee #3] | Part2.1.2 and figure1: here is a list of questions/comments which should be addressed: |
| [Referee #3] | - how are the valves switched? |
| [Hoheisel et al.] | We switch the valves manually, after passing a methane plume. For clarification, we now have added the phrase "*by switching the valves manually*". |
| | |
| [Referee #3] | - how is the flow measured and/or controlled? |
| [Hoheisel et al.] | The flow is adjusted by needle valves and measured by a flowmeter from time to time. |
| [Referee #3] | -why the flow presented here differs so much from the laboratory setup (160ml/min vs 20 to 80ml/min)? |
| [Hoheisel et al.] | In the laboratory we use a rotary valve to switch between cylinder and ambient air measurements and we will have a small flow rate. In the mobile setup, we would like to have a higher flow rate for a better time resolution and a shorter lag time between air sampling at inlet and measurement in the cavity of the CRDS analyser. Therefore, we bypass the rotary valve. |
| | |
| [Referee #3] | - what is the typical air flow going through the AC in monitoring mode? |
| [Hoheisel et al.] | The typical air flow through the AirCore in monitoring mode is 0.8 l/min. |
| | |
| [Referee #3] | - what is the typical vehicle speed while in monitoring mode? |
| [Hoheisel et al.] | The speed depends strongly on the road. We tried to drive as slowly as possible, but without constraining other vehicles. Typical vehicle speeds are 10-50 km/h. |
| | |
| [Referee #3] | - in figure1, the blue and green arrows are difficult to differentiate. Please, make them thicker. |
| [Hoheisel et al.] | Done. |
| | |
| [Referee #3] | Part3: how do you make sure that the direct samples are not mixed with ambient air? This could bias the isotopic signatures. |
| [Hoheisel et al.] | We take gas samples for example directly from the gas collecting system of the landfill and the WWTP. Therefore, the $CH_4$ concentration of these samples is between 50-90% and a potential mixture with small amounts of ambient air would be negligible. We now have clarified it and changed the text. "*Gas samples taken directly from different installations (e.g. natural gas pipelines, biogas plants, gas collecting systems of landfills and wastewater treatment plants) need to be diluted before the measurement with the CRDS analyser, because such samples usually consist of between 50 and 90% $CH_4$.*" |
| | |
| [Referee #3] | Part 3.2: it has been shown in figure5 that the $CH_4$ peak height above the background is mostly driving the fit error. It would make much more sense to present all the peak heights above the background instead of the absolute $CH_4$ value. |
| [Hoheisel et al.] | We generally agree with the reviewer. However, in the text we wanted to give an expression of the $CH_4$ concentrations measured around the different sources. To clarify it, we changed the phrase "peak height", when we report maximum $CH_4$ concentrations measured in the plume. Since it is correct, that the $CH_4$ peak height above the background is mostly driving the fit error, we decided, to change the maximum values reported in Table 2 and in the supplements to peak heights above baseline. |
| | |
| [Referee #3] | Conclusion: the Miller-Tans and Keeling approaches give the same results. Why do you suggest using one instead of the other one? |
| [Hoheisel et al.] | We agree that this is misleading. In our study we have seen that when using the York fit, it does not matter if we use the Miller-Tans or the Keeling approach. So we changed it in the conclusion. |
| | |
| [Referee #3] | P1, line 6: C2H6 only affects the 13C measurements. |
| [Hoheisel et al.] | Yes, we changed it. |
| | |
| [Referee #3] | P1, line 11: 13CH4 signatures instead of values. |
| [Hoheisel et al.] | Done. |

| [Referee #3] | P2, line 1: "biogas burning" is not a CH4 source. |
|---|---|
| [Hoheisel et al.] | Yes, corrected to "biomass burning". |

| [Referee #3] | P2, line 11: do you mean "due to its origin,"? |
|---|---|
| [Hoheisel et al.] | Yes, we have changed it. |

| [Referee #3] | P2, line 15: introduce IRMS here and not line 17, and add what GC stands for. |
|---|---|
| [Hoheisel et al.] | Done. |

| [Referee #3] | P2, line 16: I suggest to delete "has been"! as shown by Röckmann et al. |
|---|---|
| [Hoheisel et al.] | Done. |

| [Referee #3] | P3, line 1: replace signature by ratio. Signatures are not directly measured by the CRDS. |
|---|---|
| [Hoheisel et al.] | Corrected. |

| [Referee #3] | P3, line 13: it is surprising to observe such a flow range (factor 4). Could you explain why and give more details? Is the flow controlled somehow? |
|---|---|
| [Hoheisel et al.] | Ok, we have made several changes:
*"The gasflow to the analyser is typically 25 to 35ml/min for calibration gas, target gas and sample bag measurements. For some applications like ambient air measurements the flow is higher with values around 80ml/min to resolve shorter temporal variabilities. The flow is measured by an electronic flow meter (model: 5067-0223, Agilent Technologies, Inc., Santa Clara, CA) before entering the analyser."*
Tests have shown that in the flow regime of 25-80ml/min the measurement did not depend on the flow. An electronic flow meter measures the flow but the flow is not controlled. |

| [Referee #3] | P3, line 17: what do you mean by synthetic air? Have you checked it is CH4 free? |
|---|---|
| [Hoheisel et al.] | We now have added the composition (*20.5±0.5% O2 in N2*). We had also checked that it is $CH_4$ free. |

| [Referee #3] | P3, line 20: do you keep the 15min or is there a stabilization time? |
|---|---|
| [Hoheisel et al.] | Yes, there is a stabilisation time, so we cut off the first 5 minutes. We have included the following sentence.
*"However, only the last 10min were used to take into account the stabilisation time."* |

| [Referee #3] | P3, line 25: decabon. |
|---|---|
| [Hoheisel et al.] | Done. |

| [Referee #3] | P3, line 33: 160ml/min to be consistent with the previous part. |
|---|---|
| [Hoheisel et al.] | Changed as suggested. |

| [Referee #3] | P4, line 5: How the measurement precision can be better in replay mode than in monitoring mode with the same instrument? Please clarify? |
|---|---|
| [Hoheisel et al.] | The measurement precision of the analyser is the same in replay and in monitoring mode. What we wanted to point out is that in replay mode we have a higher time resolution and so more data points to describe the peak. The higher amount of data points also leads to a higher precision of the determined isotopic source signature. Since the sentence is misleading, we removed the phrase "*and a better precision*". |

| [Referee #3] | P4, line 30: have you tested the Nafion for potential fractionation? |
|---|---|
| [Hoheisel et al.] | Yes, our tests did not show a fractionation. |

| [Referee #3] | P5, line 29: Assan et al. 2017 showed that the intercept changes in time due to baseline drift. Have you regularly checked it? |
|---|---|
| [Hoheisel et al.] | During the testing phase the intercept stayed constant. |

| [Referee #3] | P5, line 33: please, check the unit. |
| [Hoheisel et al.] | Corrected. |

[Referee #3]     P6, line 1: what do you mean by fully?
[Hoheisel et al.]     We replaced the phrase "fully" with "yet".

[Referee #3]     P6, line 6: you cannot get a concentration range with a single cylinder. Please reformulate.
[Hoheisel et al.]     Yes, we changed it.
"*The gas cylinder used for calibration was chosen according to the experiment to ensure a similar composition and similar concentrations for sample and standard*".

[Referee #3]     P6, line 8: as I understood, a one point calibration strategy is used, meaning that you assume that the instrument has a linear response through the all measurement scale (mobile and sample measurements). Then why two different cylinders are used as calibration gases?
[Hoheisel et al.]     We only used one calibration cylinder for each calibration. We use a cylinders for samples with atmospheric $CH_4$ concentrations and a different one for samples around 10ppm $CH_4$ since we have noticed that the instrument drift in $\delta^{13}CH_4$ is stronger for lower $CH_4$ concentrations.

[Referee #3]     P6, line 10: have you seen some changes in the CRDS regime before and after each experiments? What was the maximal drift observed and how are they taking into account?
[Hoheisel et al.]     The maximal drift of the $CH_4$ concentration was around 0.3ppb. We now added the phrase:
"*To take into account possible drifts during the measurement we determined the time function of the standard ($\delta^{13}CH_{4\ Standard}$), used in the one point calibration, for each measuring point with a linear interpolation between the two calibration measurements.*"

[Referee #3]     P6, line 4 to 11: it is not clear to me how the CH4 is calibrated. There is no CH4 calibration factor in Table1.
[Hoheisel et al.]     Yes, we have changed the text for better understanding and we added a short explanation according the one point calibration.
"*All data measured with the CRDS analyser in the laboratory or during mobile campaigns was corrected using the factors from Table 1 and following Fig.3 prior to the one point calibration calculation.*"
"*The gas cylinder used for calibration was chosen according to the experiment to ensure a similar composition and similar concentrations for sample and standard.*"

[Referee #3]     P6, line 25: Please, detail how these uncertainties are calculated.
[Hoheisel et al.]     We calculated these relative increase by comparing the error of $\delta^{13}CH_4$ before and after the correction and calibration. We changed the text to*:*
"*Due to the correction and calibration $\delta^{13}CH_4$ there is a relative increase in the uncertainty of $\delta^{13}CH_4$ of approximately 3 to 12% for $H_2O$ concentrations below 1.3% and atmospheric $CH_4$ concentrations.*"

[Referee #3]     P6, line 32: is it the last 10min over the 15min measurements?
[Hoheisel et al.]     Yes, corrected.

[Referee #3]     P7, line 5: is it a linear drift? The uncertainties is larger than the drift itself.
[Hoheisel et al.]     Yes, because the uncertainties are larger than the drift itself we did not make a drift correction. We only use it to have an estimate how long a sample can be stored in the sample bag and in addition to quantify if a bag is leaky.

[Referee #3]     P7, line 19: please clarify which uncertainties you are talking about.
[Hoheisel et al.]     Ok, we have corrected it to *"fit uncertainties".*

[Referee #3]     P7, line 23/24: were you driving while analyzing the AC? Micro-vibrations due to vehicle motion degrade the CRDS performances.

| | |
|---|---|
| [Hoheisel et al.] | No, the vehicle stands while analysing the AC. We also noticed that especially the measurement of $C_2H_6$ is not as good as in the laboratory. |
| | |
| [Referee #3] | P7, line 25: do you mean uncertainties? Is that calculated only from the fits? |
| [Hoheisel et al.] | Yes, we changed it to *"fit uncertainty"*. |
| | |
| [Referee #3] | P8, line 17: isotopic signatures are not directly measured. |
| [Hoheisel et al.] | Yes, we have changed it to: *"the isotopic signatures of $CH_4$ from the AirCore measurements"*. |
| | |
| [Referee #3] | P8, line 33: do you mean the fit error? Or is there more parameters used to derived the uncertainties? |
| [Hoheisel et al.] | Yes, we meant fit error. Corrected. |
| | |
| [Referee #3] | P9, line 10: what precision? |
| [Hoheisel et al.] | Clarified. We added *"the precision of the determined $\delta^{13}CH_4$ signature"*. |
| | |
| [Referee #3] | P10, line 17/18: these criteria are already described earlier, no need to state it again here. |
| [Hoheisel et al.] | Ok. We still keep the criteria, because we find them importance enough to remind them again in this context. |
| | |
| [Referee #3] | P10, line 20: why is the daily mean used and not the single signatures? Same p13 line 6. |
| [Hoheisel et al.] | We discuss the single signatures as well as the averaged daily mean values for each source. Because we had taken a different number of AirCore measurements per day and per site, the simple average over all samples can be biased compared to the average over the daily mean values. |
| | By changing the phrase in P10 line 20 (see below) and including the phrase *"averaged daily mean"*, we try to make it easier to understand in the. |
| | *"During each measurement day one to five AirCore measurements were carried out at selected $CH_4$ sources. In addition to the single signatures the averaged daily mean isotopic signatures of each $CH_4$ source were calculated (see Fig.9, Table 2, and Supplement TableS1)"*. |
| | |
| [Referee #3] | P11, line 30: what is the peak height of the third AC? Please replaced value with signature. |
| [Hoheisel et al.] | Done. The three peak heights are 8.3, 8.5 and 8.9ppm. |
| | |
| [Referee #3] | P14, line 10; replace the dot with and. |
| [Hoheisel et al.] | Done. |
| | |
| [Referee #3] | P14, line 11: I suggest to replace monitor by sample. |
| [Hoheisel et al.] | Yes, corrected. |
| | |
| [Referee #3] | P14, line 15: choose between "always" and "mostly". |
| [Hoheisel et al.] | OK, I removed both. |
| | |
| [Referee #3] | P14, line 16: only peak heights are measured, not fluxes. I would then delete "therefore" and add "from these natural gas facilities". |
| [Hoheisel et al.] | Yes, corrected. |
| | |
| [Referee #3] | P15, line 3: what is the detection limit of the system? Are you sure there is plumes coming out? |
| [Hoheisel et al.] | Ok, we changed the text. |
| | *"Around the opencast mine Hambach, the $CH_4$ concentration varied only slightly between 1.94 and 1.98 ppm when we measured upwind as well as downwind of the pit. Therefore, it was not possible to identify an emission peak."* |
| | |
| [Referee #3] | P15, line 6: check for typo. |
| [Hoheisel et al.] | Done. |

[Figure]

**Figure 7.** Isotopic signature and $C_2H_6$ to $CH_4$ ratio of gas samples from a biogas plant, a landfill, a WWTP and the natural gas distribution system in Heidelberg measured between end of 2016 and November 2018.

---

## Author Response (AR2)

Dear Thomas Röckmann,

Thank you for your helpful comments. We replaced concentration by mole fraction and updated the reference to the non-peer-reviewed article.

On behalf of all co-authors, Best regards,

Antje Hoheisel

We would like to thank the reviewer for the helpful comments and suggestions. We answer each of them here after, with the original comment in blue and our response in black. We added the modifications done in the revised version in italics.

| Specific Comment                  | S:                                                                                                                                                                                                                                                                                                                                                                                                                                                                                                                                                                                                                                                                                                                                                                                                                                                                                                                                                                                                                                                                                                                                                                                                                                                                                                                                                                                                                                                                                                                                                                                                                                                                                                                                                                                                                 |
|-----------------------------------|--------------------------------------------------------------------------------------------------------------------------------------------------------------------------------------------------------------------------------------------------------------------------------------------------------------------------------------------------------------------------------------------------------------------------------------------------------------------------------------------------------------------------------------------------------------------------------------------------------------------------------------------------------------------------------------------------------------------------------------------------------------------------------------------------------------------------------------------------------------------------------------------------------------------------------------------------------------------------------------------------------------------------------------------------------------------------------------------------------------------------------------------------------------------------------------------------------------------------------------------------------------------------------------------------------------------------------------------------------------------------------------------------------------------------------------------------------------------------------------------------------------------------------------------------------------------------------------------------------------------------------------------------------------------------------------------------------------------------------------------------------------------------------------------------------------------|
| [Referee #1]
[Hoheisel et al.] | One clear omission is the $C_2H_6$ / $CH_4$ signatures from the individual plume
measurements. A great deal of important work is done in calibrating the C2H6
channel from the instrument, but then the field data from this channel is not
presented. Reporting these ratios would serve to validate that the instrument is
reporting reasonable values; conversely, if the ratios do not make sense (e.g., if the
landfill C2/C1 ratio is not zero within uncertainties), then that may point to an
additional unknown systematic error in the system. For example, these data could
be easily added to Table 2.
Similarly, for the bag samples of the local natural gas source, the measurements of
the ethane content would be an interesting addition to the manuscript.
Furthermore, the ethane content in the natural gas infrastructure is generally
recorded by the natural gas distribution company; the values obtained from the
analysis could be compared to the known values.
Thank you for pointing this out, we have added an additional section (see below)                                                                                                                                                                                                                                                                                                                                                                                                                                                                                                                                                                                                                                                                                                |
|                                   | and another figure.
" $C_2H_6$ to $CH_4$ ratio of direct samples and mobile measurements
The $C_2H_6$ to $CH_4$ ratio of gas samples from the natural gas distribution network in
Heidelberg varies between 0.027 and 0.072 with lower values in winter, 0.04±0.01,
and higher ones in summer, 0.06 ± 0.01. This result can indicate that the
percentage of Russian gas is higher in winter than in summer taking into account,
that Nitschke-Kowsky et al., (2012) reported a $C_2H_6$ to $CH_4$ ratio for Russian natural
gas to be 0.014 while for North Sea gas it is 0.078. Also the isotopic signatures of
our natural gas samples support this trend with more depleted values in winter than
in summer.
Gas emitted by other $CH_4$ sources like landfills, biogas plants and wastewater
treatment plants do not contain $C_2H_6$ . The $C_2H_6$ to $CH_4$ ratio of the gas samples
taken directly at the gas collecting systems of these sources are zero within the
errors and can be clearly separated from the natural gas samples (see Fig.7).
The separation due to $C_2H_6$ to $CH_4$ ratio works well with direct gas samples, but
unfortunately not yet for mobile AirCore measurements. In contrast to the direct
sample measurement, in mobile AirCore measurements $CH_4$ and $C_2H_6$ emitted by
the source are diluted in the background. To determine the ratio, a linear fit of the
measured $C_2H_6$ concentration to the measured $CH_4$ concentration is used.
However, due to the high uncertainty of the $C_2H_6$ measurement without averaging
as it is possible for the direct sample measurement, in combination with the very
small or not existing changes in $C_2H_6$ the fit does not provide reasonable data." |
| [Referee #1]                      | A second aspect of the manuscript that could use more attention is the calibration
of d13C-CH4. Hoheisel (2017) is cited, but at least a brief description should be
provided here (and/or a more detailed description in the supplemental material).
For example, there are terms in Fig 3 (e.g., dCH4_Nominal) which are not even
described in the text. This is critical information if these data are to be used in
partitioning of global or regional methane emissions. It is not clear from the text or
the figure whether standards with a range of isotope ratios was used in the
calibration: please clarify.                                                                                                                                                                                                                                                                                                                                                                                                                                                                                                                                                                                                                                                                                                                                                                                                                                                                                                                                                                                                                                                                                                                                                                       |
| [Hoheisel et al.]                 | We understand the point of the referee. For a better understanding, we deleted "and calibration" on page 4 line 22, since in this section we focus on corrections. In addition, we added some explanations in section 2.3.3 and in the caption of Figure 3. This section also provides more information about the used standard. "All data measured with the CRDS analyser in the laboratory or during mobile campaigns was corrected using the factors from Table 1 and following Fig.3 prior to the one point calibration calculation." "The gas cylinder used for calibration was chosen according to the experiment to ensure a similar composition and similar concentrations for sample and standard." "Scheme to correct and calibrate $C_2H_6$ and $\delta^{13}CH_4$ . $\delta^{13}CH_4$ Nominal is the nominal (known value of the standard) and $\delta^{13}CH_4$ Standard the measured (or interpolated) value of the calibration standard."                                                                                                                                                                                                                                                                                                                                                                                                                                                                                                                                                                                                                                                                                                                                                                                                                                                            |

| Detailed Comments and Questions:  |                                                                                                                                                                                                                                                                                                                                                                                                                                                                                                                                               |  |
|-----------------------------------|-----------------------------------------------------------------------------------------------------------------------------------------------------------------------------------------------------------------------------------------------------------------------------------------------------------------------------------------------------------------------------------------------------------------------------------------------------------------------------------------------------------------------------------------------|--|
| [Referee #1]                      | P1L3: "Therefore" poor word usage. Rewrite.                                                                                                                                                                                                                                                                                                                                                                                                                                                                                                   |  |
| [Honeisei et al.]                 | Agreed, we have changed it.                                                                                                                                                                                                                                                                                                                                                                                                                                                                                                                   |  |
| [Referee #1]
[Hoheisel et al.] | P3L17: polypropylene
Yes, corrected.                                                                                                                                                                                                                                                                                                                                                                                                                                                                                                       |  |
| [Referee #1]
[Hoheisel et al.] | P3L27: 260 amp-hr. check units
Done.                                                                                                                                                                                                                                                                                                                                                                                                                                                                                                       |  |
| [Referee #1]
[Hoheisel et al.] | P3L28: Supplier & Model Number for the inverter
Done, we now have included the Supplier and model number:
" e-ast HighPowerSinus HPLS 1000-12 ".                                                                                                                                                                                                                                                                                                                                                                                 |  |
| [Referee #1]
[Hoheisel et al.] | P4L3: How were the valves triggered for "replay" mode? Add a description.
We switch the valves manually after passing a methane plume. For clarification, we
now have added the following phrase.
" by switching the valves manually ".                                                                                                                                                                                                                                                                                       |  |
| [Referee #1]                      | P6L24-25: Relative increase of which uncertainty? This statement is unclear.                                                                                                                                                                                                                                                                                                                                                                                                                                                                  |  |
| [Hoheisel et al.]                 | Ok, we have changed it.
"the uncertainty of $\delta^{13}CH_4$ "                                                                                                                                                                                                                                                                                                                                                                                                                                                                     |  |
| [Referee #1]
[Hoheisel et al.] | P7L29: Please provide a brief description of the York fit.
Thank you for pointing this out. We have now included a short description.
"York's solution is the general LSE (least-squares estimation) solution: his
equations provide the best possible, unbiased estimates of the true intercept a and
slope b in all cases where the points are independent and the errors are normally
distributed" (Wehr and Saleska, 2017)                                                                                                 |  |
| [Referee #1]                      | P8L12: Why do the Keeling and MT methods produce identical source signatures and uncertainties (as shown in the figure)? I assume that this is an error in the                                                                                                                                                                                                                                                                                                                                                                                |  |
| [Hoheisel et al.]                 | preparation of the figure. If not, please explain.
We appreciate the recommendation for clarity and we looked again into the matter.
However, with both methods (KP and MT) the isotopic signature is determined
identical within the first few decimal places when using the measured data and also
for simulated datasets.                                                                                                                                                                                                      |  |
| [Referee #1]                      | P10L13: It's surprising to me that north sea gas is high in ethane content and heavy in isotope ratio. Generally heavy isotope ratios are associated with low C2+                                                                                                                                                                                                                                                                                                                                                                             |  |
| [Hoheisel et al.]                 | We have checked it again and the values are correct.                                                                                                                                                                                                                                                                                                                                                                                                                                                                                          |  |
| [Referee #1]                      | P11L33 (for example). The plume peak heights should be reported above the local baseline value, rather than absolute methane values. Please clarify in the description (e.g., "Peak height above baseline")                                                                                                                                                                                                                                                                                                                                   |  |
| [Hoheisel et al.]                 | We generally agree with the reviewer. However, in the text we wanted to give an expression of the CH 4 concentrations measured around the different sources. To clarify it, we change the phrase "peak height", when we report maximum CH 4 concentrations measured in the plume. Since it is correct, that the CH 4 peak heights above the background are mostly driving the fit error, we decided, to change the maximum values reported in Table 2 and in the supplements to peak heights above baseline. |  |

| [Referee #1]      | P12 Landfill section: The isotope signature of landfills are complicated by a) the isotope signature of the source material in the landfill, this fractionation that occurs by the anaerobic bacteria, and the further fractionation that can occur in the landfill cap where the methane is consumed aerobically.                                                                                                                                                                                                                                                                                |
|-------------------|---------------------------------------------------------------------------------------------------------------------------------------------------------------------------------------------------------------------------------------------------------------------------------------------------------------------------------------------------------------------------------------------------------------------------------------------------------------------------------------------------------------------------------------------------------------------------------------------------|
| [Hoheisel et al.] | We now have added an additional sentence at the beginning of the section.
"In addition to the source material and fractionation during the production the
isotopic signature of gas emitted by landfills depends also on fractionation
processes in the upper soil layers of the landfill. Due to the presence of oxygen in
the upper soil layers, aerobic bacteria oxidate parts of CH 4 which diffuses through
the soil cover and shift the isotopic composition to more depleted values.
Bergamaschi et al. (1998) measured these different isotopic signatures." |
| [Referee #1]      | P13: WWTP: The methane emissions from WWTPs can come from various locations and processes within the facility, and these processes may have different isotope signatures. Further, different WWTP plants employ different techniques to digest the waste; some discussion of these processes and their likely impact on your measurements would be helpful                                                                                                                                                                                                                                        |
| [Hoheisel et al.] | We understand the concern of the referee and added a few minor explanations (see below). However, more details about the specific processes lead too far for this rather technical paper. Indeed, the study will be continued and another paper is planned that will describe the different processes in more detail. "There, the sludge treatment inside the digestion towers takes place in three septic tanks with a volume of 2500m3 each under anaerobic mesophilic conditions, that means without oxygen at 37°C."                                                        |

We would like to thank the reviewer for the helpful comments and suggestions. We answer each of them here after, with the original comment in blue and our response in black. We added the modifications done in the revised version in italics.

| [Referee #2]
[Hoheisel et al.] | I suggest altering the title to something like "Improved method for mobile
characterisation of 13CH4 source signatures and its application in Germany." The
current title sells short of a major contribution of the paper.
Thank you for pointing this out. We have changed the title.    |
|-----------------------------------|-----------------------------------------------------------------------------------------------------------------------------------------------------------------------------------------------------------------------------------------------------------------------------------------------------|
| Detailed comment
[Referee #2]  | ts follow:
1. P 1, In 3: Remove "Therefore"; This sentence doesn't logically follow from the                                                                                                                                                                                                     |
| [Hoheisel et al.]                 | Yes, corrected.                                                                                                                                                                                                                                                                                     |
| [Referee #2]
[Hoheisel et al.] | 2. P 1, In 5: Explain gas matrix or replace with less jargon.
Ok, we have changed it.
"To achieve precise results a CRDS analyser,, was characterised especially with regard to cross sensitivities of composition differences of the gas matrix in air samples or calibration tanks". |
| [Referee #2]                      | 3. P 1, In 7: Abundant in many natural gases, but not all. Dry gas regions can                                                                                                                                                                                                                      |
| [Hoheisel et al.]                 | Yes, we have changed it.
" $C_2H_6$ is typically abundant in natural gases".                                                                                                                                                                                                                     |
| [Referee #2]                      | 4. P 1, In 16: A 2.8 per mil difference begs the question whether the above                                                                                                                                                                                                                         |
| [Hoheisel et al.]                 | In the 1990s Levin et al. (1999) measured $\delta^{13}CH_4$ with a mass spectrometer after separation of CH 4 from other components of air and so there is no cross-sensitivity with C 2 H 6 .                                                                     |
| [Referee #2]                      | 5. Switching between "CH4" and "methane" throughout the MS. Check for                                                                                                                                                                                                                               |
| [Hoheisel et al.]                 | Thanks, "methane" has been replaced in the manuscript by "CH 4 ".                                                                                                                                                                                                                        |
| [Referee #2]
[Hoheisel et al.] | 6. P 1, In 20: Use original data references instead of reviews, e.g., https://agupubs.onlinelibrary.wiley.com/doi/full/10.1029/2009GL039780 Yes, corrected.                                                                                                                                         |
| [Referee #2]
[Hoheisel et al.] | 7. P 2, In 1: You probably mean biomass burning.
Thanks, corrected.                                                                                                                                                                                                                              |
| [Referee #2]                      | 8. P 2, In 2: More accurate to say "e.g. a sub-category of fossil fuel extraction".                                                                                                                                                                                                                 |
| [Hoheisel et al.]                 | Clarified.                                                                                                                                                                                                                                                                                          |
| [Referee #2]                      | 9. P 2, In 7: Do you mean to say that the isotopic signal was used to diagnose                                                                                                                                                                                                                      |
| [Hoheisel et al.]                 | Agreed, we have clarified it.                                                                                                                                                                                                                                                                       |
| [Referee #2]
[Hoheisel et al.] | 10. P 2, In 13: What do you mean with "all of these seasonal variations"? Yes, this was unclear. We wanted to write " seasonal variations ".                                                                                                                                                 |
| [Referee #2]
[Hoheisel et al.] | 11. P 2, In 26: Briefly describe the importance/use of a storage tube (first mention). Done.                                                                                                                                                                                                        |
| [Referee #2]                      | 12. P 2, In 26: Nine facilities in 21 campaigns? Do you mean 21 measurement                                                                                                                                                                                                                         |
| [Hoheisel et al.]                 | Yes, we have changed it to "21 mobile measurement days".                                                                                                                                                                                                                                            |

| [Referee #2]                      | 13. P 3, In 15: Perhaps that's explained later, but how do you get so close to the                                                                                                                                                                                                                                                                                                                                                                                                                                                                                                                                                                                                                                                                                                                                                                                                                                                                                                      |
|-----------------------------------|-----------------------------------------------------------------------------------------------------------------------------------------------------------------------------------------------------------------------------------------------------------------------------------------------------------------------------------------------------------------------------------------------------------------------------------------------------------------------------------------------------------------------------------------------------------------------------------------------------------------------------------------------------------------------------------------------------------------------------------------------------------------------------------------------------------------------------------------------------------------------------------------------------------------------------------------------------------------------------------------|
| [Hoheisel et al.]                 | We take gas samples for example directly from the gas collecting system of the landfill and the WWTP. We now have clarified it and changed the phrase.
"Gas samples taken directly from different installations (e.g. natural gas pipelines, biogas plants, gas collecting systems of landfills and wastewater treatment plants) need to be diluted before the measurement with the CRDS analyser, because such samples usually consist of between 50 and 90% CH 4 ."                                                                                                                                                                                                                                                                                                                                                                                                                                                                                                     |
| [Referee #2]
[Hoheisel et al.] | 14. P 3, In 17: Specify the composition of synthetic air. We now have added the composition ( $20.5\pm0.5\%$ $O_2$ in $N_2$ ).                                                                                                                                                                                                                                                                                                                                                                                                                                                                                                                                                                                                                                                                                                                                                                                                                                                          |
| [Referee #2]                      | 15. P 3, In 33: Do you mean a 20-25 sec lag time between air sampling at inlet and fully arriving in the cavity? Sounds like it, but not fully clear.                                                                                                                                                                                                                                                                                                                                                                                                                                                                                                                                                                                                                                                                                                                                                                                                                                   |
| [Hoheisel et al.]                 | Yes, we clarified it.
"Due to the length of the intake line, the volume of the cavity, and a flow rate of
160ml/min the time lag between air sampling at inlet and measurement in the
cavity of the CRDS analyser is 20 to 25sec."                                                                                                                                                                                                                                                                                                                                                                                                                                                                                                                                                                                                                                                                                                                                             |
| [Referee #2]                      | 16. P 4, In 11: What do you mean with representations have the same width? They don't. It's a bit confusing. The next sentence already states that both peaks represent the same emission plume.                                                                                                                                                                                                                                                                                                                                                                                                                                                                                                                                                                                                                                                                                                                                                                                        |
| [Hoheisel et al.]                 | What we meant is that when measuring a CH 4 plume first without and subsequently with the AirCore the two measured CH 4 peaks have the same height, but since we measured with a slower flow in monitoring mode not the same width. For better comparison we stretched the peak measured without the AirCore in x direction, to make it easier to compare the measurements done with and without the AirCore and to note that both peaks have the same shape and height. We also changed the text to: "For comparison the peak measured in `monitoring mode' (blue dots/line on the left side) is stretched by a factor of 12.5 in x-direction (blue line on the right side) so that the peak measured with the AirCore and the stretched one measured without it have the same width. The peak measured in `replay mode' precisely corresponds to the stretched one measured in `monitoring mode', because both peaks reproduce the same emission plume." |
| [Referee #2]                      | 17. P 4, In 13: Any hypotheses why this may be the case? This seems to be an important observation                                                                                                                                                                                                                                                                                                                                                                                                                                                                                                                                                                                                                                                                                                                                                                                                                                                                                      |
| [Hoheisel et al.]                 | As described above we measured the air stored in the AirCore directly after we measured a $CH_4$ peak in monitoring mode. So the storage time is relatively short and we expect that the peaks measured with and without the AirCore have the same shape and the same height.                                                                                                                                                                                                                                                                                                                                                                                                                                                                                                                                                                                                                                                                                                           |
| [Referee #2]                      | 18. Table 1: Is the unit for ethane sensitivity to water (ppm) a typo? Units of ppb would make more sense. For reference, atmospheric ethane background concentrations are in the order of $< 1$ ppb.                                                                                                                                                                                                                                                                                                                                                                                                                                                                                                                                                                                                                                                                                                                                                                                   |
| [Hoheisel et al.]                 | No, the unit ppm is correct.                                                                                                                                                                                                                                                                                                                                                                                                                                                                                                                                                                                                                                                                                                                                                                                                                                                                                                                                                            |
| [Referee #2]                      | 19. P 5, In 5: Natural gases on a continuum between 0 to 40% ethane have been measured. See Sherwood et al., 2017 (already in your refs). Hence, the importance of ethane correction varies: very important for wet gas basins (meaning lots of associated gas) and less important for very dry gas regions (mature thermogenic dry gas).                                                                                                                                                                                                                                                                                                                                                                                                                                                                                                                                                                                                                                               |
| [Hoheisel et al.]                 | I nank you for pointing this out. In this study we measured natural gas samples
from the natural gas network in Heidelberg. Therefore, we wanted to give an
overview of C 2 H 6 contents of natural gases in the pipeline network in Germany. We
have now added:
"As typical natural gases in the pipeline network in Germany contain"                                                                                                                                                                                                                                                                                                                                                                                                                                                                                                                                                                                                         |

| [Referee #2]
[Hoheisel et al.] | 20. P 5, In 10: where does the 3 per mil value come from?
The value of 3‰, is the bias $\delta^{13}$ CH 4 has in presence of a C 2 H 6 to CH 4 ration of 0.073, which was the highest value we measured for our natural gas samples. To make the text more understandable, we changed it.
"A correction is necessary because for typical C 2 H 6 to CH 4 ratios between 0.027 and 0.073 measured for our natural gas samples, $\delta^{13}$ CH 4 shows a bias between 1 and 3‰ to more enriched values. We must also keep in mind that similar shifts in, $\delta^{13}$ CH 4 to less enriched values can occur when using a calibration cylinder which contains C 2 H 6 ."                                                                                                                                                                                                                                                                                                                                                                                                                                                                                                                                                                                                                                                                                                            |
|-----------------------------------|----------------------------------------------------------------------------------------------------------------------------------------------------------------------------------------------------------------------------------------------------------------------------------------------------------------------------------------------------------------------------------------------------------------------------------------------------------------------------------------------------------------------------------------------------------------------------------------------------------------------------------------------------------------------------------------------------------------------------------------------------------------------------------------------------------------------------------------------------------------------------------------------------------------------------------------------------------------------------------------------------------------------------------------------------------------------------------------------------------------------------------------------------------------------------------------------------------------------------------------------------------------------------------------------------------------------------------------------------------------------------------------------------------------------------------------------------------------------------------------------------------------------|
| [Referee #2]                      | 21. P 7, In 3: What is the sign of the drift? Is the drift due to fractionation in the bag                                                                                                                                                                                                                                                                                                                                                                                                                                                                                                                                                                                                                                                                                                                                                                                                                                                                                                                                                                                                                                                                                                                                                                                                                                                                                                                                                                                                                           |
| [Hoheisel et al.]                 | or due to leakage of background air into the bag?
The sign is + and the major cause for the drift is fractionation by the leakage of
sample air out of the bag. So for a better understanding we added the phrase "to
more enriched values" and the following sentence.
"The drift occurs especially due to fractionation by diffusion of air through the
sample bag".                                                                                                                                                                                                                                                                                                                                                                                                                                                                                                                                                                                                                                                                                                                                                                                                                                                                                                                                                                                                                                                                                                                                |
| [Referee #2]
[Hoheisel et al.] | 22. P 7, In 28: What are simulated data?
How we simulated data is explained a few lines below. For better understanding we changes this section:
"Similar to the method described by Wehr and Saleska (2017) for $CO_2$ and $\delta^{13}CO_2$ , we create several typical emission plume crossings. We generated synthetic $CH_4$ peaks using a background concentration of 1.95 ppm $CH_4$ and a Gaussian curve with 10-280 equidistant data points every 3.7 s and an enhancement of 100-10000 ppb. The corresponding $\delta^{13}CH_4$ values were calculated with $CH_4$ source signatures between $\cdot 35\%$ and $-65\%$ and a background of $-48\%$ . To reproduce the statistical uncertainties of a real measurement, we add a normally distributed scattering around zero to the synthetic $CH_4$ concentrations and the corresponding isotope ratios. The standard deviation of the normal distributed scattering depends on the $CH_4$ concentrations and was chosen as the Allan standard deviation measured for raw data of the analyser. However, when simulating possible improved analysers, we reduced the scattering by a factor 2 to 10.
Such sets of data were generated 5000 times for each condition. To study the influence of the averaging time, we calculate mean data sets with varying averaging periods (up to 1 min).
For each dataset the $\delta^{13}CH_4$ source signature was calculated with the Miller-Tans and the Keeling method using the York or the OLS fit." |
| [Referee #2]                      | 23. P 10, In 4: I'm confused. Here it says no significant seasonal cycle has been observed. In the next paragraph, it says the values are more depleted in winter than in summer. Are there enough samples to determine this correctly?                                                                                                                                                                                                                                                                                                                                                                                                                                                                                                                                                                                                                                                                                                                                                                                                                                                                                                                                                                                                                                                                                                                                                                                                                                                                              |
| [Hoheisel et al.]                 | We changed this paragraph.                                                                                                                                                                                                                                                                                                                                                                                                                                                                                                                                                                                                                                                                                                                                                                                                                                                                                                                                                                                                                                                                                                                                                                                                                                                                                                                                                                                                                                                                                           |
| [Referee #2]
[Hoheisel et al.] | 24. P 10, In 13: What are the uncertainties for the ethane-to-methane ratios? The uncertainties for the ethane-to-methane ratios is in both cases 0.01. We now have included it.                                                                                                                                                                                                                                                                                                                                                                                                                                                                                                                                                                                                                                                                                                                                                                                                                                                                                                                                                                                                                                                                                                                                                                                                                                                                                                                                     |
| [Referee #2]                      | 25. P 12, In 2: Why were the plumes expected to be smaller than on the other                                                                                                                                                                                                                                                                                                                                                                                                                                                                                                                                                                                                                                                                                                                                                                                                                                                                                                                                                                                                                                                                                                                                                                                                                                                                                                                                                                                                                                         |
| [Hoheisel et al.]                 | Both farms have a different number of dairy cows. We now added the comment
"due to lower animal number".                                                                                                                                                                                                                                                                                                                                                                                                                                                                                                                                                                                                                                                                                                                                                                                                                                                                                                                                                                                                                                                                                                                                                                                                                                                                                                                                                                                                          |
| [Referee #2]
[Hoheisel et al.] | 26. P 12, In 11: Why are seasonal differences for the biogas plant expected? Thanks, we have removed "seasonal".                                                                                                                                                                                                                                                                                                                                                                                                                                                                                                                                                                                                                                                                                                                                                                                                                                                                                                                                                                                                                                                                                                                                                                                                                                                                                                                                                                                                     |
| [Referee #2]                      | 27. P 12, par. 1 & 3: Is C3/C4 diet information available for the Ladenburg and Kleve farms?                                                                                                                                                                                                                                                                                                                                                                                                                                                                                                                                                                                                                                                                                                                                                                                                                                                                                                                                                                                                                                                                                                                                                                                                                                                                                                                                                                                                                         |
| [Hoheisel et al.]                 | No, unfortunately not.                                                                                                                                                                                                                                                                                                                                                                                                                                                                                                                                                                                                                                                                                                                                                                                                                                                                                                                                                                                                                                                                                                                                                                                                                                                                                                                                                                                                                                                                                               |

| [Referee #2]
[Hoheisel et al.] | <li>28. P 14, In 11: How small the fluxes were actually depends on the size of the facility (is it possible to detect all plumes at once?) and the wind conditions in addition to the measured methane concentrations.</li><li>The facility is in a small forest and so in difficult terrain. It is unlikely that we detect all plumes at once. So we weakened our statement.</li> |
|-----------------------------------|---------------------------------------------------------------------------------------------------------------------------------------------------------------------------------------------------------------------------------------------------------------------------------------------------------------------------------------------------------------------------------------------|
| [Referee #2]                      | 29. P 14, In 20: Did your measurements include incomplete combustion from the compressor turbines? This could be detected by the associated CO2 or CO signal. It is still an open question whether high heat leads to isotopic fractionation of the source                                                                                                                                  |
| [Hoheisel et al.]                 | To determine the origin of a CH 4 plume measured around the natural gas facility between Hähnlein und Gernsheim is difficult, because this facility contains a natural gas storage and several compressor stations even operated by different gas providers. At this site further work and more measuring campaigns are planned to receive more detailed results.                |
| [Referee #2]                      | 30. P 14, In 25: Do you use "open" and "in service" interchangeably? Not clear. If yes, it's not a surprise that a mine currently in service produces more emissions than a closed mine.                                                                                                                                                                                                    |
| [Hoheisel et al.]                 | Yes, we changed "open" to "in service".                                                                                                                                                                                                                                                                                                                                                     |
| [Referee #2]                      | 31. P 14, In 30: The Bottrop mine shaft measurements do not match coal bed gas samples except for the closed mine with a large error bar and only one AirCore measurement.                                                                                                                                                                                                                  |
| [Hoheisel et al.]                 | Done.                                                                                                                                                                                                                                                                                                                                                                                       |
| [Referee #2]                      | 32. Conclusions: In the first paragraph, it's important to highlight again that these                                                                                                                                                                                                                                                                                                       |
| [Hoheisel et al.]                 | OK, we have added the following phrase to highlight your recommendation.
"characterisation of each individual analyser"                                                                                                                                                                                                                                                                  |

We would like to thank the reviewer for the helpful comments and suggestions. We answer each of them here after, with the original comment in blue and our response in black. We added the modifications done in the revised version in italics.

| General Comments                                       | 5                                                                                                                                                                                                                                                                                                                                                                                                                                                                                                                                                                                                                                                                                                                                                                                                                                                                                                                                                                                                                                                                                                                                                                                                                                                                                                                                                                                                                                                                                                                                                                                                                                                                                                                                                                                           |
|--------------------------------------------------------|---------------------------------------------------------------------------------------------------------------------------------------------------------------------------------------------------------------------------------------------------------------------------------------------------------------------------------------------------------------------------------------------------------------------------------------------------------------------------------------------------------------------------------------------------------------------------------------------------------------------------------------------------------------------------------------------------------------------------------------------------------------------------------------------------------------------------------------------------------------------------------------------------------------------------------------------------------------------------------------------------------------------------------------------------------------------------------------------------------------------------------------------------------------------------------------------------------------------------------------------------------------------------------------------------------------------------------------------------------------------------------------------------------------------------------------------------------------------------------------------------------------------------------------------------------------------------------------------------------------------------------------------------------------------------------------------------------------------------------------------------------------------------------------------|
| [Referee #3]                                           | It is nevertheless regrettable that the C2C6 to CH4 ratios are not presented along with the isotopic signatures. I strongly suggest to add it in order to give the paper more visibility.                                                                                                                                                                                                                                                                                                                                                                                                                                                                                                                                                                                                                                                                                                                                                                                                                                                                                                                                                                                                                                                                                                                                                                                                                                                                                                                                                                                                                                                                                                                                                                                                   |
| [Hoheisel et al.]                                      | We appreciate the recommendation and we added the short section $C_2H_6$ to $CH_4$ ratio of direct samples and mobile measurements as well as one Figure.                                                                                                                                                                                                                                                                                                                                                                                                                                                                                                                                                                                                                                                                                                                                                                                                                                                                                                                                                                                                                                                                                                                                                                                                                                                                                                                                                                                                                                                                                                                                                                                                                                   |
|                                                        | " $C_2H_6$ to $CH_4$ ratio of direct samples and mobile measurements
The $C_2H_6$ to $CH_4$ ratio of gas samples from the natural gas distribution network in
Heidelberg varies between 0.027 and 0.072 with lower values in winter, 0.04±0.01,
and higher ones in summer, 0.06 ± 0.01. This result can indicate that the
percentage of Russian gas is higher in winter than in summer taking into account,
that Nitschke-Kowsky et al., (2012) reported a $C_2H_6$ to $CH_4$ ratio for Russian natural
gas to be 0.014 while for North Sea gas it is 0.078. Also the isotopic signatures of
our natural gas samples support this trend with more depleted values in winter than
in summer.
Gas emitted by other $CH_4$ sources like landfills, biogas plants and wastewater
treatment plants do not contain $C_2H_6$ . The $C_2H_6$ to $CH_4$ ratio of the gas samples
taken directly at the gas collecting systems of these sources are zero within the
errors and can be clearly separated from the natural gas samples (see Fig.7).
The separation due to $C_2H_6$ to $CH_4$ ratio works well with direct gas samples, but
unfortunately not yet for mobile AirCore measurements. In contrast to the direct
sample measurement, in mobile AirCore measurements $CH_4$ and $C_2H_6$ emitted by
the source are diluted in the background. To determine the ratio, a linear fit of the
measured $C_2H_6$ concentration to the measured $CH_4$ concentration is used.
However, due to the high uncertainty of the $C_2H_6$ measurement without averaging
as it is possible for the direct sample measurement, in combination with the very
small or not existing changes in $C_2H_6$ the fit does not provide reasonable data." |
| Specific comments
[Referee #3]
[Hoheisel et al.] | The language used in the manuscript is not always precise. For instance, the terms "13C values" and "13C signatures" are not the same and sometimes confusing as used in the MS. Also, the 13C signatures are not directly measured by the CRDS but determined after data treatment from the measurements. Please correct through the all manuscript. Yes, we clarified it.                                                                                                                                                                                                                                                                                                                                                                                                                                                                                                                                                                                                                                                                                                                                                                                                                                                                                                                                                                                                                                                                                                                                                                                                                                                                                                                                                                                                                 |
| [Referee #3]                                           | Most of the time, the uncertainties are presented, but without any explanation on
how they are calculated and which parameters are used to computed them. It is
also not clear what is the difference between uncertainties and precision. For
instance, page 9 line 5/6, it is stated that increasing the number of data point
improves the uncertainties on signatures. The next sentence, "uncertainties" has
been replaced by "precision". It is confusing. Please clarify through the all
manuscript.                                                                                                                                                                                                                                                                                                                                                                                                                                                                                                                                                                                                                                                                                                                                                                                                                                                                                                                                                                                                                                                                                                                                                                                                                                                                |
| [Hoheisel et al.]                                      | Yes, we corrected it.                                                                                                                                                                                                                                                                                                                                                                                                                                                                                                                                                                                                                                                                                                                                                                                                                                                                                                                                                                                                                                                                                                                                                                                                                                                                                                                                                                                                                                                                                                                                                                                                                                                                                                                                                                       |
| [Referee #3]                                           | The G2201-i instrument can be used in different mode, which drives the measurement frequency of each species. Also, the methane is reported by the instrument in high range (HR) and high precision (HP) mode depending on the level of the measured mole fractions. This drives the instrumental precision. Please, add some details on these two points.                                                                                                                                                                                                                                                                                                                                                                                                                                                                                                                                                                                                                                                                                                                                                                                                                                                                                                                                                                                                                                                                                                                                                                                                                                                                                                                                                                                                                                  |
| [Hoheisel et al.]                                      | We agree that this is an important point and added the following text.
"All measurements with the CRDS analyser were done in the combined CO 2 /CH 4
mode to measure CH 4 and CO 2 parallel. In addition, the High Precision (HP) mode
for CH 4 is chosen to provide the most precise CH 4 measurements for CH 4
concentrations up to 12 ppm."                                                                                                                                                                                                                                                                                                                                                                                                                                                                                                                                                                                                                                                                                                                                                                                                                                                                                                                                                                                                                                                                                                                                                                                                                                                                                             |

| [Referee #3]                      | Part2.1.2 and figure1: here is a list of questions/comments which should be addressed:                                                                                                                                                                                                                                                                                                                                                                                                                                                                                                                                               |
|-----------------------------------|--------------------------------------------------------------------------------------------------------------------------------------------------------------------------------------------------------------------------------------------------------------------------------------------------------------------------------------------------------------------------------------------------------------------------------------------------------------------------------------------------------------------------------------------------------------------------------------------------------------------------------------|
| [Referee #3]                      | - how are the valves switched?                                                                                                                                                                                                                                                                                                                                                                                                                                                                                                                                                                                                       |
| [Hoheisel et al.]                 | We switch the valves manually, after passing a methane plume. For clarification, we now have added the phrase " by switching the valves manually ".                                                                                                                                                                                                                                                                                                                                                                                                                                                                           |
| [Referee #3]
[Hoheisel et al.] |  <li>how is the flow measured and/or controlled?</li> <li>The flow is adjusted by needle valves and measured by a flowmeter from time to time.</li>                                                                                                                                                                                                                                                                                                                                                                                                                                                                         |
| [Referee #3]                      | -why the flow presented here differs so much from the laboratory setup (160ml/min                                                                                                                                                                                                                                                                                                                                                                                                                                                                                                                                                    |
| [Hoheisel et al.]                 | In the laboratory we use a rotary valve to switch between cylinder and ambient air measurements and we will have a small flow rate. In the mobile setup, we would like to have a higher flow rate for a better time resolution and a shorter lag time between air sampling at inlet and measurement in the cavity of the CRDS analyser. Therefore, we bypass the rotary valve.                                                                                                                                                                                                                                                       |
| [Referee #3]
[Hoheisel et al.] | - what is the typical air flow going through the AC in monitoring mode?
The typical air flow through the AirCore in monitoring mode is 0.8 l/min.                                                                                                                                                                                                                                                                                                                                                                                                                                                                                 |
| [Referee #3]
[Hoheisel et al.] | - what is the typical vehicle speed while in monitoring mode?
The speed depends strongly on the road. We tried to drive as slowly as possible,
but without constraining other vehicles. Typical vehicle speeds are 10-50 km/h.                                                                                                                                                                                                                                                                                                                                                                                                 |
| [Referee #3]                      | - in figure1, the blue and green arrows are difficult to differentiate. Please, make                                                                                                                                                                                                                                                                                                                                                                                                                                                                                                                                                 |
| [Hoheisel et al.]                 | Done.                                                                                                                                                                                                                                                                                                                                                                                                                                                                                                                                                                                                                                |
| [Referee #3]                      | Part3: how do you make sure that the direct samples are not mixed with ambient                                                                                                                                                                                                                                                                                                                                                                                                                                                                                                                                                       |
| [Hoheisel et al.]                 | We take gas samples for example directly from the gas collecting system of the landfill and the WWTP. Therefore, the CH4 concentration of these samples is between 50-90% and a potential mixture with small amounts of ambient air would be negligible. We now have clarified it and changed the text.
"Gas samples taken directly from different installations (e.g. natural gas pipelines, biogas plants, gas collecting systems of landfills and wastewater treatment plants) need to be diluted before the measurement with the CRDS analyser, because such samples usually consist of between 50 and 90% CH 4 ." |
| [Referee #3]                      | Part 3.2: it has been shown in figure5 that the CH4 peak height above the background is mostly driving the fit error. It would make much more sense to present all the peak heights above the background instead of the absolute CH4 value.                                                                                                                                                                                                                                                                                                                                                                                          |
| [Hoheisel et al.]                 | We generally agree with the reviewer. However, in the text we wanted to give an expression of the CH 4 concentrations measured around the different sources. To clarify it, we changed the phrase "peak height", when we report maximum CH 4 concentrations measured in the plume. Since it is correct, that the CH 4 peak height above the background is mostly driving the fit error, we decided, to change the maximum values reported in Table 2 and in the supplements to peak heights above baseline.                                                                                         |
| [Referee #3]                      | Conclusion: the Miller-Tans and Keeling approaches give the same results. Why do                                                                                                                                                                                                                                                                                                                                                                                                                                                                                                                                                     |
| [Hoheisel et al.]                 | We agree that this is misleading. In our study we have seen that when using the York fit, it does not matter if we use the Miller-Tans or the Keeling approach. So we changed it in the conclusion.                                                                                                                                                                                                                                                                                                                                                                                                                                  |
| [Referee #3]
[Hoheisel et al.] | P1, line 6: C2H6 only affects the 13C measurements.
Yes, we changed it.                                                                                                                                                                                                                                                                                                                                                                                                                                                                                                                                                           |
| [Referee #3]
[Hoheisel et al.] | P1, line 11: 13CH4 signatures instead of values.
Done.                                                                                                                                                                                                                                                                                                                                                                                                                                                                                                                                                                            |

| [Referee #3]
[Hoheisel et al.] | P2, line 1: "biogas burning" is not a CH4 source.
Yes, corrected to "biomass burning".                                                                                                                                                                                                                                                                                                                                                                                                                                                                                                                                                                       |
|-----------------------------------|-----------------------------------------------------------------------------------------------------------------------------------------------------------------------------------------------------------------------------------------------------------------------------------------------------------------------------------------------------------------------------------------------------------------------------------------------------------------------------------------------------------------------------------------------------------------------------------------------------------------------------------------------------------------|
| [Referee #3]
[Hoheisel et al.] | P2, line 11: do you mean "due to its origin,"?
Yes, we have changed it.                                                                                                                                                                                                                                                                                                                                                                                                                                                                                                                                                                                      |
| [Referee #3]
[Hoheisel et al.] | P2, line 15: introduce IRMS here and not line 17, and add what GC stands for. Done.                                                                                                                                                                                                                                                                                                                                                                                                                                                                                                                                                                             |
| [Referee #3]
[Hoheisel et al.] | P2, line 16: I suggest to delete "has been"! as shown by Röckmann et al. Done.                                                                                                                                                                                                                                                                                                                                                                                                                                                                                                                                                                                  |
| [Referee #3]                      | P3, line 1: replace signature by ratio. Signatures are not directly measured by the CRDS.                                                                                                                                                                                                                                                                                                                                                                                                                                                                                                                                                                       |
| [Hoheisel et al.]                 | Corrected.                                                                                                                                                                                                                                                                                                                                                                                                                                                                                                                                                                                                                                                      |
| [Referee #3]                      | P3, line 13: it is surprising to observe such a flow range (factor 4). Could you explain why and give more details? Is the flow controlled somehow?                                                                                                                                                                                                                                                                                                                                                                                                                                                                                                             |
| [Hoheisel et al.]                 | Ok, we have made several changes:
"The gasflow to the analyser is typically 25 to 35ml/min for calibration gas, target
gas and sample bag measurements. For some applications like ambient air
measurements the flow is higher with values around 80ml/min to resolve shorter
temporal variabilities. The flow is measured by an electronic flow meter (model:
5067-0223, Agilent Technologies, Inc., Santa Clara, CA) before entering the
analyser."
Tests have shown that in the flow regime of 25-80ml/min the measurement did not
depend on the flow. An electronic flow meter measures the flow but the flow is not
controlled. |
| [Referee #3]
[Hoheisel et al.] | P3, line 17: what do you mean by synthetic air? Have you checked it is CH4 free? We now have added the composition ( $20.5\pm0.5\%$ O2 in N2). We had also checked that it is CH 4 free.                                                                                                                                                                                                                                                                                                                                                                                                                                                             |
| [Referee #3]
[Hoheisel et al.] | P3, line 20: do you keep the 15min or is there a stabilization time?
Yes, there is a stabilisation time, so we cut off the first 5 minutes. We have
included the following sentence.
"However, only the last 10min were used to take into account the stabilisation
time."                                                                                                                                                                                                                                                                                                                                                                  |
| [Referee #3]
[Hoheisel et al.] | P3, line 25: decabon.
Done.                                                                                                                                                                                                                                                                                                                                                                                                                                                                                                                                                                                                                                  |
| [Referee #3]
[Hoheisel et al.] | P3, line 33: 160ml/min to be consistent with the previous part.
Changed as suggested.                                                                                                                                                                                                                                                                                                                                                                                                                                                                                                                                                                        |
| [Referee #3]                      | P4, line 5: How the measurement precision can be better in replay mode than in monitoring mode with the same instrument? Please clarify?                                                                                                                                                                                                                                                                                                                                                                                                                                                                                                                        |
| [Hoheisel et al.]                 | The measurement precision of the analyser is the same in replay and in monitoring mode. What we wanted to point out is that in replay mode we have a higher time resolution and so more data points to describe the peak. The higher amount of data points also leads to a higher precision of the determined isotopic source signature. Since the sentence is misleading, we removed the phrase "and a better precision".                                                                                                                                                                                                                                      |
| [Referee #3]
[Hoheisel et al.] | P4, line 30: have you tested the Nafion for potential fractionation?
Yes, our tests did not show a fractionation.                                                                                                                                                                                                                                                                                                                                                                                                                                                                                                                                            |
| [Referee #3]                      | P5, line 29: Assan et al. 2017 showed that the intercept changes in time due to baseline drift. Have you regularly checked it?                                                                                                                                                                                                                                                                                                                                                                                                                                                                                                                                  |
| [Hoheisel et al.]                 | During the testing phase the intercept stayed constant.                                                                                                                                                                                                                                                                                                                                                                                                                                                                                                                                                                                                         |

| [Referee #3]
[Hoheisel et al.] | P5, line 33: please, check the unit.
Corrected.                                                                                                                                                                                                                                                                                                                                                                                                                                                                                          |
|-----------------------------------|---------------------------------------------------------------------------------------------------------------------------------------------------------------------------------------------------------------------------------------------------------------------------------------------------------------------------------------------------------------------------------------------------------------------------------------------------------------------------------------------------------------------------------------------|
| [Referee #3]
[Hoheisel et al.] | P6, line 1: what do you mean by fully?
We replaced the phrase "fully" with "yet".                                                                                                                                                                                                                                                                                                                                                                                                                                                        |
| [Referee #3]                      | P6, line 6: you cannot get a concentration range with a single cylinder. Please                                                                                                                                                                                                                                                                                                                                                                                                                                                             |
| [Hoheisel et al.]                 | Yes, we changed it.
"The gas cylinder used for calibration was chosen according to the experiment to
ensure a similar composition and similar concentrations for sample and standard".                                                                                                                                                                                                                                                                                                                                                |
| [Referee #3]                      | P6, line 8: as I understood, a one point calibration strategy is used, meaning that you assume that the instrument has a linear response through the all measurement scale (mobile and sample measurements). Then why two different cylinders are                                                                                                                                                                                                                                                                                           |
| [Hoheisel et al.]                 | We only used one calibration cylinder for each calibration. We use a cylinders for samples with atmospheric CH 4 concentrations and a different one for samples around 10ppm CH 4 since we have noticed that the instrument drift in $\delta^{13}CH_4$ is stronger for lower CH 4 concentrations.                                                                                                                                                                                                          |
| [Referee #3]                      | P6, line 10: have you seen some changes in the CRDS regime before and after each experiments? What was the maximal drift observed and how are they taking into account?                                                                                                                                                                                                                                                                                                                                                                     |
| [Hoheisel et al.]                 | The maximal drift of the CH 4 concentration was around 0.3ppb. We now added the phrase:
"To take into account possible drifts during the measurement we determined the time function of the standard ( $\delta^{13}$ CH 4 Standard ), used in the one point calibration, for each measuring point with a linear interpolation between the two calibration measurements."                                                                                                                                |
| [Referee #3]                      | P6, line 4 to 11: it is not clear to me how the CH4 is calibrated. There is no CH4                                                                                                                                                                                                                                                                                                                                                                                                                                                          |
| [Hoheisel et al.]                 | Yes, we have changed the text for better understanding and we added a short
explanation according the one point calibration.
"All data measured with the CRDS analyser in the laboratory or during mobile
campaigns was corrected using the factors from Table 1 and following Fig.3 prior to
the one point calibration calculation."
"The gas cylinder used for calibration was chosen according to the experiment to
ensure a similar composition and similar concentrations for sample and standard." |
| [Referee #3]
[Hoheisel et al.] | P6, line 25: Please, detail how these uncertainties are calculated.
We calculated these relative increase by comparing the error of $\delta^{13}CH_4$ before and after the correction and calibration. We changed the text to:
"Due to the correction and calibration $\delta^{13}CH_4$ there is a relative increase in the uncertainty of $\delta^{13}CH_4$ of approximately 3 to 12% for H 2 O concentrations below 1.3% and atmospheric CH 4 concentrations."                                                |
| [Referee #3]
[Hoheisel et al.] | P6, line 32: is it the last 10min over the 15min measurements?
Yes, corrected.                                                                                                                                                                                                                                                                                                                                                                                                                                                           |
| [Referee #3]
[Hoheisel et al.] | P7, line 5: is it a linear drift? The uncertainties is larger than the drift itself.
Yes, because the uncertainties are larger than the drift itself we did not make a drift
correction. We only use it to have an estimate how long a sample can be stored in
the sample bag and in addition to quantify if a bag is leaky.                                                                                                                                                                                                       |
| [Referee #3]
[Hoheisel et al.] | P7, line 19: please clarify which uncertainties you are talking about.
Ok, we have corrected it to "fit uncertainties" .                                                                                                                                                                                                                                                                                                                                                                                                          |
| [Referee #3]                      | P7, line 23/24: were you driving while analyzing the AC? Micro-vibrations due to vehicle motion degrade the CRDS performances.                                                                                                                                                                                                                                                                                                                                                                                                              |

| [Hoheisel et al.]                 | No, the vehicle stands while analysing the AC. We also noticed that especially the measurement of $C_2H_6$ is not as good as in the laboratory.                                                                                                                                                                                                                                                                                                                                                                                                                                                                                                                                                                                                                        |
|-----------------------------------|------------------------------------------------------------------------------------------------------------------------------------------------------------------------------------------------------------------------------------------------------------------------------------------------------------------------------------------------------------------------------------------------------------------------------------------------------------------------------------------------------------------------------------------------------------------------------------------------------------------------------------------------------------------------------------------------------------------------------------------------------------------------|
| [Referee #3]
[Hoheisel et al.] | P7, line 25: do you mean uncertainties? Is that calculated only from the fits? Yes, we changed it to "fit uncertainty".                                                                                                                                                                                                                                                                                                                                                                                                                                                                                                                                                                                                                                         |
| [Referee #3]
[Hoheisel et al.] | P8, line 17: isotopic signatures are not directly measured.
Yes, we have changed it to: "the isotopic signatures of CH 4 from the AirCore measurements" .                                                                                                                                                                                                                                                                                                                                                                                                                                                                                                                                                                                  |
| [Referee #3]                      | P8, line 33: do you mean the fit error? Or is there more parameters used to derived                                                                                                                                                                                                                                                                                                                                                                                                                                                                                                                                                                                                                                                                                    |
| [Hoheisel et al.]                 | Yes, we meant fit error. Corrected.                                                                                                                                                                                                                                                                                                                                                                                                                                                                                                                                                                                                                                                                                                                                    |
| [Referee #3]
[Hoheisel et al.] | P9, line 10: what precision?
Clarified. We added "the precision of the determined $\delta^{13}CH_4$ signature".                                                                                                                                                                                                                                                                                                                                                                                                                                                                                                                                                                                                                                              |
| [Referee #3]                      | P10, line 17/18: these criteria are already described earlier, no need to state it again here                                                                                                                                                                                                                                                                                                                                                                                                                                                                                                                                                                                                                                                                          |
| [Hoheisel et al.]                 | Ok. We still keep the criteria, because we find them importance enough to remind them again in this context.                                                                                                                                                                                                                                                                                                                                                                                                                                                                                                                                                                                                                                                           |
| [Referee #3]                      | P10, line 20: why is the daily mean used and not the single signatures? Same p13 line 6                                                                                                                                                                                                                                                                                                                                                                                                                                                                                                                                                                                                                                                                                |
| [Hoheisel et al.]                 | We discuss the single signatures as well as the averaged daily mean values for
each source. Because we had taken a different number of AirCore measurements
per day and per site, the simple average over all samples can be biased compared
to the average over the daily mean values.
By changing the phrase in P10 line 20 (see below) and including the phrase
"averaged daily mean", we try to make it easier to understand in the.
"During each measurement day one to five AirCore measurements were carried
out at selected CH 4 sources. In addition to the single signatures the averaged daily
mean isotopic signatures of each CH 4 source were calculated (see Fig.9, Table 2,
and Supplement TableS1)". |
| [Referee #3]                      | P11, line 30: what is the peak height of the third AC? Please replaced value with signature.                                                                                                                                                                                                                                                                                                                                                                                                                                                                                                                                                                                                                                                                           |
| [Hoheisel et al.]                 | Done. The three peak heights are 8.3, 8.5 and 8.9ppm.                                                                                                                                                                                                                                                                                                                                                                                                                                                                                                                                                                                                                                                                                                                  |
| [Referee #3]
[Hoheisel et al.] | P14, line 10; replace the dot with and.
Done.                                                                                                                                                                                                                                                                                                                                                                                                                                                                                                                                                                                                                                                                                                                       |
| [Referee #3]
[Hoheisel et al.] | P14, line 11: I suggest to replace monitor by sample.
Yes, corrected.                                                                                                                                                                                                                                                                                                                                                                                                                                                                                                                                                                                                                                                                                               |
| [Referee #3]
[Hoheisel et al.] | P14, line 15: choose between "always" and "mostly".
OK, I removed both.                                                                                                                                                                                                                                                                                                                                                                                                                                                                                                                                                                                                                                                                                             |
| [Referee #3]                      | P14, line 16: only peak heights are measured, not fluxes. I would then delete "therefore" and add "from these natural gas facilities"                                                                                                                                                                                                                                                                                                                                                                                                                                                                                                                                                                                                                                  |
| [Hoheisel et al.]                 | Yes, corrected.                                                                                                                                                                                                                                                                                                                                                                                                                                                                                                                                                                                                                                                                                                                                                        |
| [Referee #3]                      | P15, line 3: what is the detection limit of the system? Are you sure there is plumes                                                                                                                                                                                                                                                                                                                                                                                                                                                                                                                                                                                                                                                                                   |
| [Hoheisel et al.]                 | Coming out?
Ok, we changed the text.
"Around the opencast mine Hambach, the CH 4 concentration varied only slightly
between 1.94 and 1.98 ppm when we measured upwind as well as downwind of
the pit. Therefore, it was not possible to identify an emission peak."                                                                                                                                                                                                                                                                                                                                                                                                                                                                             |
| [Referee #3]
[Hoheisel et al.] | P15, line 6: check for typo.
Done.                                                                                                                                                                                                                                                                                                                                                                                                                                                                                                                                                                                                                                                                                                                                  |

**Characterisation Improved method for mobile characterisation of** $\delta^{13}$ CH4 source signatures from methane sources and its application in Germanyusing mobile measurements**

Antje Hoheisel1, Christiane Yeman1,2, Florian Dinger1,3, Henrik Eckhardt1, and Martina Schmidt1

1Institute of Environmental Physics, Heidelberg University, Heidelberg, Germany 2Laboratory of Ion Beam Physics, ETH Zurich, Zurich, Switzerland (now) 3Max-Planck Institute for Chemistry, Mainz, Germany

**Correspondence:** Antje Hoheisel (antje.hoheisel@iup.uni-heidelberg.de)

**Abstract.**

The carbon isotopic signature ( $\delta^{13}$ CH4) of several methane sources in Germany (around Heidelberg and in North Rhine-Westphalia) were characterised. Therefore, mobile Mobile measurements of the plume of CH4 sources are carried out using a cavity ring-down spectrometer (CRDS). To achieve precise results a CRDS analyser, which measures methane (CH4), car-

- 5 bon dioxide (CO2) and their 13C to 12C ratios, was characterised especially with regard to cross sensitivities of composition differences of the gas matrix in air samples or calibration tanks. The two most important gases which affect the measurements  $\delta^{13}$ CH4 are water vapour (H2O) and ethane (C2H6). To avoid the cross sensitivity with H2O, the air is dried with a nafion dryer during mobile measurements. C2H6 is typically abundant in natural gas gases and thus in methane plumes or samples originating from natural gas. A C2H6 correction and calibration are essential to obtain accurate  $\delta^{13}$ CH4 results, which can
- 10 deviate up to 3% depending on whether an ethane a  $C_2H_6$  correction is applied.

The isotopic signature is determined with the Miller-Tans approach and the York fitting method. During 21 field campaigns the mean  $\delta^{13}$ CH4 values signatures of three dairy farms (-63.9±0.9%0), a biogas plant (-62.4±1.2%0), a landfill (-58.7±3.3%0), a wastewater treatment plant (-52.5±1.4%0), an active deep coal mine (-56.0±2.3%0) and two natural gas storage and gas compressor stations (-46.1±0.8%0) were recorded.

In addition, between December 2016 and June November 2018 gas samples from the Heidelberg natural gas distribution network were measured were measured with a mean  $\delta^{13}$ CH4 value of  $-43.3 \pm 0.8\%$ . Contrary to former measurements between 1991 and 1996 (Levin et al., 1999) by Levin et al. (1999) no strong seasonal cycle is shown. The mean  $\delta^{13}$ CH4 value of this study is  $-43.1 \pm 0.8\%$  which is 2.8% more depleted than in former years.

**1 Introduction**

20 Methane (CH4) is the second most important anthropogenic greenhouse gas. The atmospheric growth rate of  $\frac{\text{CH}_4}{\text{methane}}$  has changed significantly during the last decades, stabilising at zero growth from 1999 to 2006 before beginning to increase again

after 2007 (?)(Dlugokencky et al., 2009). Several studies have focused on the recent  $CH_4$  growth caused by changes in sources and sinks (Rigby et al., 2017; Turner et al., 2017).

Recent studies by Schaefer et al. (2016), Rice et al. (2016) and Nisbet et al. (2016) have shown how the  $\delta^{13}$ CH4 measurements can help to understand the changes in global CH4 increase rates, and to assign the related source types. The stable carbon

[revised manuscript text omitted]
 CH4 and CO2 on  $\delta^{13}$ CH4 were detected up to concentrations-mole fractions of 10 ppm CH4 or rather 450 ppm CO2.

Previous studies from Rella et al. (2015) and Assan et al. (2017) have reported higher  $\delta^{13}$ CH4 results when the gas sample

- 20 contains  $C_2H_6$ . As natural gas contains As typical natural gases in the pipeline network in Germany contain between 1.4 to 7 Mol% of  $C_2H_6$  (Nitschke-Kowsky et al., 2012), the  $C_2H_6$  interference is especially important when analysing CH4 emissions from natural gas facilities or the isotopic composition of natural gas. The  $C_2H_6$  interference on  $\delta^{13}CH_4$  measurements was carefully tested with our analyser by carrying out three dilution tests, to determine a correction (Fig. S4).  $\delta^{13}CH_4$  increases linearly with increasing  $C_2H_6$  to CH4 ratio. The slope of the regression line and thus the correction factor was found to be
- 25  $40.87\pm0.49\%$  (ppm CH4)/(ppm C2H6). The A correction is necessary due to because for typical C2H6 to CH4 ratios between 0.027 and 0.073 measured for our natural gas samples,  $\delta^{13}$ CH4 showing a bias of up to between 1 and 3% in our study depending on the to more enriched values. We must also keep in mind that similar shifts in  $\delta^{13}$ CH4 to less enriched values can occur when using a calibration cylinder which contains C2H6ratio of the sample and the calibration cylinder.

**2.2.2 Correcting the measured C2H6 concentrationmole fraction**

30 To correct for the strong cross sensitivity between  $C_2H_6$  and  $\delta^{13}CH_4$  measurements, an accurate determination of the  $C_2H_6$ concentration mole fraction is required. Because the measurement of  $C_2H_6$  is an additional feature of the instrument a correction and calibration of the  $C_2H_6$  concentration mole fraction were performed. The  $C_2H_6$  concentration mole fraction decreases strongly with increasing humidity, even for  $H_2O$  concentrations mole fractions below 0.15% (Fig. S1). For humidity below 0.15% a correction factor of  $0.43\pm0.51$  (ppm  $C_2H_6$ )/(%  $H_2O$ ) was determined and for humidity higher than 0.16% the correction factor is  $0.70\pm0.10$  (ppm  $C_2H_6$ )/(%  $H_2O$ ). There is no correction for  $H_2O$  mole fractions between 0.15 and 0.16%, because in this range the behaviour of  $C_2H_6$  in the presence of  $H_2O$  changes. However, no discontinuity, such that observed by Assan et al. (2017), was seen.

Besides H2O also the concentrations mole fractions of CH4 and CO2 interfere with the measured C2H6. To determine the cross sensitivities of CH4 and CO2 on C2H6 two dilution series and three injection tests were performed and produced gas mixtures with concentration mole fraction ranges of 1.8 to 10 ppm CH4 or 2 to 600 ppm CO2. All dilution and injection tests with C2H6 concentrations mole fractions between 0 to 1.3 ppm show similar results with an average of  $0.0077 \pm 0.0007 (\text{ppm C}_2\text{H}_6)/(\text{ppm CH}_4)$  and  $(1.25 \pm 0.94) 10^{-4} (\text{ppm C}_2\text{H}_6)/(\text{ppm CO}_2)$  (Fig. S5).

To calibrate the  $C_2H_6$  measurement two dilution tests with  $C_2H_6$  concentrations mole fractions ranging from 0 to 3 ppm

were performed (Fig. S6). The measured  $C_2H_6$  concentrations mole fractions were nearly twice as large as expected. After correcting the measured  $C_2H_6$  concentrations mole fractions due to  $H_2O$ ,  $CH_4$  and  $CO_2$  a calibration factor (slope of the regression line) of  $0.538 \pm 0.002$  ppm/ppm and a calibration intercept of  $0.070 \pm 0.005$  ppm was determined.

**15 2.2.3 Calibration to international scales**

5

10

All calibration gases used in this study are compressed air filled in aluminium cylinders. The CH4 and CO2 concentrations mole fractions were calibrated against the WMO scale (Dlugokencky et al., 2005) using a GC system (Levin et al., 1999). To determine the  $\delta^{13}$ CH4 values, flasks filled from our calibration gases were sent to analysed at MPI Jena ( $\delta^{13}$ CH4: ±0.05 ppm‰). These analyses connect our Heidelberg measurements to the VPDB (Vienna Pee Dee Belemnite) isotope scale (Sperlich et al., 2016). C. H. is not followed by the state of the s

20 al., 2016).  $C_2H_6$  is not fully yet calibrated to an international scale. One calibration cylinder filled by Deuste-Steininger (Mühlhausen, Germany) with 4.98 ppm  $C_2H_6$  is certified by this company with an uncertainty of  $\pm 2\%$ .

All data measured with the CRDS analyser in the laboratory or during mobile campaigns was corrected <del>prior to the one point calibration calculation</del> using the factors from Table 1 and following Fig. 3 - prior to the one point calibration calculation.

The gas cylinder used for calibration was chosen according to the experiment to ensure a similar composition and <del>concentration</del> range similar mole fractions for sample and standard. For ambient air measurements in the laboratory and for mobile measure-

- 25 range similar mole fractions for sample and standard. For ambient air measurements in the laboratory and for mobile measurements a gas cylinder filled with compressed air is used to calibrate the data. For diluted gas samples from  $CH_4$  sources a gas cylinder with atmospheric concentrations mole fractions spiked with natural gas to 10 ppm  $CH_4$  is used. The calibration gas is measured before and after every experiment/field campaign in the laboratory or in the vehicle. Tests at the beginning of this study showed that measurements of the calibration gas inside the vehicle do not increase the precision and are therefore
- 30 not necessary for mobile measurements of less than 10 hours. To take into account possible drifts during the measurement we determined the time function of the standard ( $\delta^{13}$ CH4standard), used in the one point calibration, for each measuring point with a linear interpolation between the two calibration measurements.

**2.2.4 Instrument performance and uncertainties**

The repeatability of the analyser as a function of the CH4 concentration mole fraction was determined by the measurement of three different gas cylinders for 120 min each. The Allan variance (Werle et al., 1993) was calculated with the raw data for averaging times of up to 11 min (Fig. 4). The Allan standard deviation  $\sigma$  (the square root of the Allan variance  $\sigma^2$ ) for the

5 raw (3.7 sec) CH4 data is between 0.34 to 2.69 ppb for gases with a CH4 concentration mole fraction of 1900 to 10000 ppb. For the corresponding  $\delta^{13}$ CH4 data, an improvement of the Allan standard deviation with higher CH4 concentration mole fraction from 3.76 to 0.77% can be seen. The Allan standard deviation of C2H6 is approximately 0.09 ppm for gases with C2H6 concentrations mole fractions up to 5 ppm.

During mobile measurements especially CH4 and  $\delta^{13}$ CH4 show rapid changes when driving through the emission plume

- 10 of a CH4 source and thus do not allow us to average the data over long time periods. However, for sample measurements in the laboratory (e.g. natural gas samples) longer averaging times of up to 10 or 15 min significantly decrease the Allan standard deviation (see Fig. 4). For a 10 min averaging period the Allan standard deviation of 1900 ppb or 10000 ppb CH4 decreases to values of 0.09 ppb and 0.47 ppb, and for  $\delta^{13}$ CH4 to values of 0.40% and 0.06%. The Allan standard deviation of C2H6 decreases to 0.006 ppm. Due to the correction and calibration of  $\delta^{13}$ CH4 there is a relative increase in the uncertainty of
- 15 approximately  $5 \delta^{13}$ CH4 of approximately 3 to 12% for H2O mole fractions below 1.3% and atmospheric CH4 mole fractions.

**2.3 Analysis of $\delta^{13}$ CH4**

20

**2.3.1 Gas samples from natural gas distribution network**

Between December 2016 and June-November 2018, gas samples from the Heidelberg natural gas distribution network were collected two to three times a month from the gas-glass blowing workshop at the university campus in one litre sample bags (Tedlar® with Polypropylen polypropylene valve with septum, Restek GmbH, Bad Homburg, Germany).

The gas samples were measured as described in Sect. 2.1.1, corrected by the factors given in Table 1 and calibrated as described above. For each gas sample the average and standard deviation of the last 10min over the 15min measurement were calculated.

To determine the repeatability of a measurement as well as the storage effect, pair samples were taken and storage tests carried out, with storage times of the bags up to 226 days and two to five measurements taken from each sample bag. Duplicate samples taken on the same day and measured one after another show a mean difference in  $\delta^{13}$ CH4 of 0.12±0.08‰ with a maximal maximum difference of 0.30‰. Storage tests of 12 natural gas samples stored on average for 104 days (41 to 226 days) in Tedlar® bags show an average drift of 0.0023±0.0028‰/ day -to more enriched values. The drift occurs especially due to fractionation by diffusion of air through the sample bag.

Since the samples are measured for the first time on average 26 days (0 to 88 days) after the sample day, the  $\delta^{13}$ CH4 signature of the samples-value of a sample will change by approximately 0.06% due to this storage in Tedlar® bags. Even after 100 days the average drift is only 0.23% and therefore for each sample the  $\delta^{13}$ CH4 values measured within 100 days after sampling were averaged. To quantify the short-term variations of  $\delta^{13}$ CH4 from the local gas supply network within one week, two samples per day\_daily gas samples were taken over 5 days at the end of November 2017 and averaged the  $\delta^{13}$ CH4 values for the duplicate samples. The maximum difference between the five averaged values samples was  $0.7 \pm 0.2\%$ .

**2.3.2 Determination of $\delta^{13}$ CH4 source signatures from mobile plume measurements**

For mobile measurements the CRDS analyser is installed inside a vehicle and measurements are carried out as described in

- 5 Sect. 2.1.2. The  $\delta^{13}$ CH4 signature of the CH4 sources were determined by the Miller-Tans approach (Miller and Tans, 2003) using the unaveraged data measured in 'replay mode' with the AirCore. To fit a linear regression line to the data the York fit (York et al., 2004) was used as recommended also by Wehr and Saleska (2017). York's solution is the general least-squares estimation solution, providing the best possible, unbiased estimates of the true intercept and slope in all cases where the points are independent and the errors are normally distributed (Wehr and Saleska, 2017). Because the York fit allows errors in x and
- 10 y, it also account for the finding that the analyser can measure  $\delta^{13}$ CH4 more accurately at higher CH4 concentrationsmole fractions. The errors for CH4 and  $\delta^{13}$ CH4 for different concentrations mole fractions were determined with the Allan standard deviation.

For accurate results the following criteria are used to select 79 AirCore measurements out of 135. Only  $\delta^{13}$ CH4 signatures with fit uncertainties lower than 5% are used. The number of data points and especially the peak height above background

- 15 concentration control the precision mole fraction control the uncertainty of the determined isotopic signature when applying a Miller-Tans Plot, therefore only plume measurements with peak heights above background concentration mole fraction higher than 0.45 ppm and more than 25 data points fulfil this criterion. Furthermore, in some cases the reported  $C_2H_6$  concentration mole fraction jumps while driving although there cannot be a change in the  $C_2H_6$  concentration mole fraction of the ambient air. These jumps in  $C_2H_6$  also results in  $\delta^{13}CH_4$  jumps. Therefore, all AirCore measurements with a sudden change in  $C_2H_6$
- 20 larger than 1 ppm were neglected. With these criteria the isotopic signature of a  $CH_4$  source determined from one AirCore plume measurement has an average precision fit uncertainty of  $1.8 \pm 1.3\%$ .

**2.3.3 Comparison of different methods to determine $\delta^{13}$ CH4 source signatures**

In order to define the optimal method for the determination of the source signature the 135 AirCore measurements as well as simulated data were used. In the following the differences in the δ13CH4 source signature when using the Keeling method or
the Miller-Tans approach (Keeling, 1958; Miller and Tans, 2003) will be discussed and the York fit will be compared to the ordinary least squares (OLS) fit (here the lm() fit function from GNU R is used).

Similar to the method described by Wehr and Saleska (2017) for CO2 and  $\delta^{13}$ CO2, we simulated create several typical emission plume crossingswith CH4 source signatures of -35% to -65% and a background of -48%. In addition, the CH4 concentration enhancements in the plume  $\Delta c_{source}$  (100 – 10000 ppb), the number of measured data points during plume

30 crossing n (10 – 280) and the averaging times (up to 1 min) were varied. For each set of conditions ( $\delta^{13}$ CH4source,  $\Delta c_{source}$ , n), we we generated synthetic CH4 concentrations and calculated the corresponding  $\delta^{13}$ CH4 values peaks using a background concentration mole fraction of 1.95 ppm CH4 
[revised manuscript text omitted]